



# Analysis of the Surface Mass Balance for Deglacial Climate Simulations

Marie-Luise Kapsch[1], Uwe Mikolajewicz[1], Florian Andreas Ziemen[1, *], Christian B. Rodehacke[2, 3], and Clemens Schannwell[1]

[1]Max Planck Institute for Meteorology, Bundesstraße 53, 20146 Hamburg, Germany.
[2]Danish Meteorological Institute, Lyngbyvej 100, 2100 Copenhagen Ø, Denmark
[3]Alfred Wegener Institut, Helmholtz Centre for Polar and Marine Research, Am Handelshafen 12, 27570 Bremerhaven, Germany.
[*]now at Deutsches Klimarechenzentrum, Bundesstr. 45a, 20146 Hamburg, Germany.

**Correspondence:** Marie-Luise Kapsch (marie.kapsch@mpimet.mpg.de)

**Abstract.** Most studies analyzing changes in the surface mass balance (SMB) of the Greenland ice sheet are limited to the last century, due to the availability of observations and the computational limitations of regional climate modeling. Using transient simulations with a comprehensive Earth System Model (ESM) we extend previous research and study changes in the SMB and equilibrium line altitude (ELA) for deglacial climate conditions. An energy balance model (EBM) is used to
downscale atmospheric processes. It determines the SMB on higher spatial resolution and allows to resolve SMB variations due to topographic gradients not resolved by the ESM. An evaluation for historical climate conditions (1980–2010) shows that derived SMBs compare well with SMBs from regional modeling. Throughout the deglaciation changes in insolation dominate the Greenland SMB: 1) The increase in insolation and associated warming early in the deglaciation result in an ELA and SMB increase. The SMB increase is caused by compensating effects of melt and accumulation, as a warmer atmosphere precipitates
more. After 13 ka before present (BP) melt begins to dominate and the SMB decreases. 2) The decline in insolation after 9 ka BP leads to an increasing SMB and decreasing ELA. Superimposed on these long-term changes are episodes of significant SMB/ELA decreases, related to slowdowns of the Atlantic Meridional Overturning Circulation (AMOC) that lead to cooling over most of the Northern Hemisphere. To study associated changes in the ice sheet geometry, the SMB data set is made available to the ice sheet modeling community.

## 1   Introduction

Mass changes of ice sheets are controlled by variations in the surface mass balance (SMB) and ice discharge (van den Broeke et al., 2009; Khan et al., 2015). The SMB is determined by mass gain due to accumulation, as a result of snow deposition, and mass loss by ablation, induced by thermodynamical processes at the surface and subsequent melt-water runoff (Ettema et al,



2009). Further, iceberg calving and basal melting at the ice-ocean and ice-bedrock interfaces impact the mass balance of ice sheets.

To model the SMB, atmospheric processes associated with the energy balance at the surface as well as snow processes, such as albedo evolution or refreezing, need to be simulated realistically (Vizcaino et al., 2014). This is specifically challenging for long-term climate simulations with state-of-the-art Earth System Models (ESMs), as the horizontal resolution is often not

sufficient to capture small scale climate features, e.g., sharp topographic gradients at the ice sheet margins. Therefore, most of the analyses on changes and variability of the SMB have been based on observations, statistical regression and correction techniques as well as simulations with high-resolution regional climate models and covered the last century only (e.g., Fettweis et al., 2008; Ettema et al, 2009; Hanna et al., 2011; Box, 2013; Fettweis et al., 2017; Noël et al., 2018).

This study extends the analysis of SMB changes to the last deglaciation (21 thousand years (ka) before present (BP) to

the present). The last deglaciation was characterized by significant changes in insolation and associated changes in ice sheets, greenhouse gas concentrations, and other amplifying feedbacks (Clark et al., 2012). The large ice loss resulted in the disappearance of the North American and Eurasian ice sheets. In the Northern Hemisphere, only the Greenland ice sheet remains at present. The retreat of the ice sheets during the deglaciation resulted in about 1 m sea-level rise per 100 years, a rate, which on average is comparable to future projections of sea-level rise (e.g., Horton et al., 2014). Superimposed on these overall changes

were periods of abrupt climate events. Some of the most prominent events are Heinrich event 1 (HE1; about 16.8 ka BP) (e.g., McManus et al., 2004; Stanford et al., 2011) and the Younger Dryas (about 13-11.5 ka BP)(Carlson et al., 2007), both associated with a major Northern Hemispheric cooling and a significant decrease of the Atlantic Overturning Circulation.

To explore the SMB under such climate conditions, transient simulations of the last deglaciation with a comprehensive ESM are used in combination with an energy balance model (EBM) to calculate the SMB. An EBM approach is computationally

more expensive as compared to other models (e.g., positive degree day models)(e.g., Tarasov and Peltier, 2002; Abe-Ouchi et al., 2007), as EBMs account for the energy balance at the surface, including snow processes, such as albedo evolution or refreezing. Hence, they mainly have been mainly for shorter-term simulations such as future predictions (Fettweis, 2007; Fettweis et al., 2013; Ettema et al, 2009; Mikolajewicz et al., 2007b; Vizcaino et al., 2010, 2015). However, using different approaches to calculate the SMB in simulations of the last glacial cycle with an intermediate complexity model, Bauer and

Ganopolski (2017) have shown that only the EBM approach results in a realistic representation of ice volume changes.

The main aim of this paper is to introduce the EBM and apply it to long term climate simulations. First, we introduce the EBM and the underlying simulations with the Max Planck Institute ESM (MPI-ESM). Second, we provide a thorough evaluation of the model performance for present-day climate conditions over Greenland, by comparing the derived SMB data set to SMBs from regional climate modeling. We then present and investigate SMB changes and variability throughout the last

deglaciation and point out mechanisms behind the SMB changes. As the SMB is a key parameter in controlling changes in the geometry of the ice sheets, this data set will be made available to the ice-sheet modeling community.



## 2   Model Systems and Data

To obtain SMB fields from long-term climate simulations the coupled MPI-ESM is used in combination with a EBM. Two kind of simulations were performed: 1) A set of historical simulations (1980–2010) to evaluate the EBM derived SMBs. For this, we force the EBM with output from historical simulations with MPI-ESM as well as ERA-Interim reanalysis and compare the obtained SMBs to SMBs derived with the regional climate model MAR (Modèle Atmosphérique Régional) (Fettweis, 2007); 2) Simulations of the last deglaciation with prescribed ice sheet boundary conditions to investigate SMB changes under transient climate forcing. The simulations performed for this study are summarized in Table 2.

### 2.1   The Surface Energy and Mass Balance Model

We use the EBM to downscale the SMB from the coarse resolution atmospheric model grid onto high-resolution ice-sheet topographies. The main challenge in downscaling the SMB is to realistically capture the small scale features of both melt and accumulation. Melt and accumulation are highly dependent on the topographic height, e.g., at a given time, low elevations might experience melt while higher elevations remain frozen. Projecting melt and accumulation on a topography with better resolved vertical gradients has therefore a significant impact on the SMB. Differences between the original and downscaled SMBs are therefore mainly a result of differences in the elevation rather than the horizontal grid refinement. To account for this, we employ a 3-D EBM scheme that is forced with high-frequency atmospheric data. The EBM scheme is an enhanced version of the energy and mass balance code that has been used to couple the ice sheet model SICOPOLIS to a previous version of MPI-ESM (Mikolajewicz et al., 2007b; Vizcaino et al., 2008, 2010). The main improvements are 1) an advanced broadband albedo scheme considering aging, snow depth dependency, and the influence of the cloud coverage, 2) the consideration of vertical movement of the snow/ice properties and compaction, 3) rain-induced change of the heat content of the snow layers, and 4) an enhanced refreezing scheme. We further adopted the scheme by introducing elevation classes. They were first introduced by Vizcaino et al. (2013) within the CESM-Glimmer model setup and have the advantage that the model becomes computationally cheaper, as the SMB is computed on the native and coarse resolution atmospheric grid instead of the high-resolution ice sheet topography. The obtained 3-D fields can be interpolated onto different ice sheet topographies (see Section 2.3). In the following, we present the basic structure of the EBM, including its improvements as compared to the scheme used in Vizcaino et al. (2010).

### Height correction

To compute the SMB, atmospheric fields are mapped onto 24 fixed elevation levels, ranging from sea level to 8000 m (we use irregular intervals that start with 100 m distance at the surface and increase with height). To account for height differences between each of these elevation classes and the surface elevation of the atmospheric model, a height correction is applied to near-surface air temperature, humidity, dew point, precipitation, downward longwave radiation, and near-surface density fields. The downward shortwave radiation is kept constant, as it is largely affected by atmospheric properties independent of elevation differences (e.g., ozone concentration, aerosol thickness) (Yang et al., 2006).

The following height corrections are applied before the EBM calculations:





- Precipitation rates are corrected under consideration of the height-desertification effect. This halves the precipitation for
  an orography height difference of 1000 m above a threshold height of 2000 m for each grid point (Budd and Smith,
  1979).

- Near-surface air temperature and dew point are corrected using a constant lapse-rate of -4.6 K km$^{-1}$, similar to the value
  proposed in Abe-Ouchi et al. (2007). The specific humidity, which can be used alternatively to the dew point temperature
  to calculate the latent heat flux (Bolton, 1980), is decreased with height, under the assumption that the relative humidity
  stays constant throughout the atmospheric column.

- The surface pressure is adjusted under the assumption of a typical atmospheric density/pressure profile $p = p_{\mathrm{atm0}} \exp\left(\frac{-z}{H_s}\right)$,
  where $p_{\mathrm{atm0}}$ is the pressure at the surface, z is the height and $H_s$ the scale height with a typical value of 8.4 km.

- The downward longwave radiation is corrected by applying the observed constant radiation gradient of Marty et al.
  (2002) and is reduced by 29 W m$^{-2}$ km$^{-1}$ (see also Wild et al., 1995).

**Surface mass and energy balance calculation**

Accumulation and melt determine the SMB. Precipitation is accumulated and falls as snow with a density of 300 kg m$^{-3}$ when
the height-corrected near-surface air temperature is lower than the freezing temperature of 273.15 K. Otherwise, precipitation
falls as rain.

The computation of melt requires a snow/ice model, as the melt rate depends on the heat content of snow and the heat
exchange between the surface snow layer and the atmosphere above as well as the snow/ice layers below. To account for this,
the EBM consists of a 5-layer snow model, discretized into layers of increasing thickness. The model considers only vertical
exchanges because the horizontal extent is several orders of magnitude larger than the vertical extent. The snow model starts
initially with a reference density describing the typical exponentially increasing density with depth (Cuffey, 2010). The top
layer's exchange with the atmosphere is computed from short- and longwave radiation fluxes, latent and sensible heat fluxes,
and the heat release due to the immediate refreezing of rain, if the surface layer has a temperature below the freezing temper-
ature. After updating the heat content of the surface layer, the temperature difference with the layer below determines the heat
flux into the layer below by taking an ice/snow density-dependent heat conductivity into account (Fukusako, 1990). The heat
conductivity is a function of density following Schwerdtfeger (1963), where the conductivity of ice is $K_{\mathrm{ice}} = 2.10$ W m$^{-1}$ K$^{-1}$,
a common value within the reported range from 2.09 W m$^{-1}$ K$^{-1}$ to 2.26 W m$^{-1}$ K$^{-1}$ (Yen, 1981). We neglect any temperature
dependence of the ice and snow conductivity (Fukusako, 1990). The scheme progresses downward until the lowest layer is up-
dated. The heat flux beyond the lowest layer is assumed to be zero, in agreement with observations showing that the ice/snow
layer temperature below 10 m follows the long-term trend and not the seasonal cycle. If the ice/snow layer's temperature ex-
ceeds the melting temperature, the temperature is set to the freezing point, and the related temperature excess converts the
corresponding amount of ice/snow into liquid water. Liquid water penetrates into the layer below and refreezes in this layer as
long as the layer is colder than the freezing temperature. Any remaining liquid water may penetrate further downward, where
it potentially refreezes. Liquid water that leaves the lowest layer or flows into a layer with a density exceeding the pore close





density of 830 kg m$^{-3}$ (Pfeffer et al., 1991) is treated as run-off. Rain that precipitates on the surface refreezes in the surface layer by releasing latent heat. It increases the surface temperature until the surface reaches the melting point, which terminates refreezing. Henceforth, rain percolates into the layer below, where it is treated as meltwater.

***Albedo***

The amount of incoming radiation, which is available for heating the snow/ice layers and eventually melting, is controlled by the surface albedo $\alpha$. The albedo parametrization used here differs from Vizcaino et al. (2010). We have developed a frequency-independent broadband albedo that combines existing parameterizations, as described in this section. It represents processes neglected in conventional parameterizations and covers a broader range of albedos suggested by observational accounts.

Freshly fallen snow has (in our scheme) an albedo of 0.92 ($\alpha_{frsnow}$, see Table 1). Snow metamorphosis processes that change the snow's characteristics through the growth of larger crystals at the expense of smaller ones, ultimately transform snow into firn (Cuffey, 2010). As a result the snow albedo ($\alpha_{snow}(t)$) decreases and approaches the albedo of firn ($\alpha_{firn}$), which is parameterized by a time-dependent exponential decay (Klok and Oerlemans, 2004; Oerlemans and Knap, 1998) as

$$\alpha_{snow}(t) = \alpha_{firn} + (\alpha_{frsnow} - \alpha_{firn})\exp(-t_{snow}\hat{\tau}_a), \tag{1}$$

where $t_{snow}$ is the time since the last snowfall and $\hat{\tau}_a$ a time constant (see Table 1). This process is here referred to as "aging". Besides, the depth of the top snow layer ($d_{snow}(t)$) determines how much of a potentially darker background shines through and modulates the surface albedo (Klok and Oerlemans, 2004; Oerlemans and Knap, 1998). The equation

$$\alpha_{surface}(t, d_{snow}) = \alpha_{snow}(t) + (\alpha_{bg} - \alpha_{snow}(t))\exp(d_{snow}/\hat{d}), \tag{2}$$

renders this process, where $\alpha_{bg}$ is the background albedo, which is generally the albedo of ice, and $\hat{d}$ is $0.0024\,\mathrm{m}^{-1}$. In the
albedo parameterization melting reduces the snow thickness while snowfall increases the thickness when the precipitation rate exceeds $7.23\cdot10^{-10}\,\mathrm{m(WE)\,s}^{-1} \approx 2.5\,\mathrm{cm(WE)\,year}^{-1}$; all depths presented here are water equivalent (WE) depths (reference density of $1000\,\mathrm{kg\,m}^{-3}$). The maximum snow depth is set to 2 m WE, which corresponds to approximately 6 m of snow. Any additional snow is still considered in the layer model to close the mass calculation but it does not impact the albedo (the snow depth is an internal diagnostic variable).

Depending on the snow depth, melting and refreezing have different albedo values. When the surface experiences melt, the albedo drops to the melt albedo of snow or ice, respectively. When the surface refreezes, the albedos brighten, and the aging process starts. Compared to snow processes, the albedo differences for refrozen surfaces are smaller, and the process is slower ($\hat{\tau}_{ar}$ for refrozen snow and ice, Table 1). Depending on the snow depth, their albedos start from the refrozen values of snow ($\alpha_{snow:refrz}$) or ice ($\alpha_{ice:refrz}$). Only melted surfaces and the background do not experience any aging.

Moreover, the background albedo shows a slight density dependence, which impacts regions of persistent high melting and lowers the surface albedo via the background albedo (Eq 2). The background albedo is

$$\alpha_{bg} = \min(\alpha_{ice}, q_1\rho + q_2), \tag{3}$$





where $q_1$=-4·10$^{-4}$ m$^3$kg$^{-1}$ and $q_2$=0.95, which is similar to values published by Liston et al. (1999); see also Suzuki et al. (2006) as another example for this kind of parameterization.

Furthermore, varying cloud cover affects the surface albedo (Greuell et al., 1994) because a higher cloud cover reflects more thermal radiation downward, which shifts the broadband albedo towards lower values (darker surface). We use the linear function of Wouter et al. (1994) so that the maximum albedo change is 0.1 between a complete overcast sky and cloud-free conditions.

All albedo values used in this study (e.g., for refrozen snow/ice, fresh snow, ice, firn) are tuned for a realistic representation
of the SMB for historical climate conditions (see Section 3) and are listed in Table 1.

*Vertical advection and density evolution*

Adding snowfall and releasing meltwater drives the movement of mass through the column, which, ultimately, drives the compaction of snow. In contrast to commonly used equations that diagnose snow compaction, we apply a simple parameterization. By construction, it reproduces a defined density depth profile perfectly if snow is added to the snow column at the top while
melting is absent. Our reference density profile increases exponentially with depth (Cuffey, 2010). Parallel to the movement of mass, the corresponding temperature profile also shifts.

Our scheme handles two cases: 1) where surface melting subtracts mass from the top layer or 2) where snowfall adds mass at the top. The inflow of snow and ice from above is added to each layer by increasing the layer's density until it reaches the reference density of this layer. Any additional inflow from above bypasses the layer and is redirected into the layer below.
Once mass flows out of the bottom layer, it leaves the system. In the case of surface melting, the entire density profile is lifted upward by the corresponding amount. An inflow of mass through the bottom layer closes the mass budget. The inflow has a density of a virtual layer beneath the bottom box.

In each layer, the entry of freshwater from above refreezes if the layer's temperature is colder than the freezing point temperature, as described above. Water refreezes as ice with a density of 917 kg m$^{-3}$. As a consequence, the layer's density
grows and eventually exceeds the layer's reference density; here, it occurs without triggering any mass exchange beyond the layer. Although the mass entering the lowest layer from below is lower than the density of ice, in ablation areas, sustained refreezing increases gradually the density of layers by refrozen freshwater. Once the pore closure density is reached, it prevents further percolation into deeper layers.

## 2.2    The Max Planck Institute Earth System Model

The simulations in this study are performed with the Max Planck Institute for Meteorology Earth System Model (MPI-ESM, version 1.2; see Mikolajewicz et al., 2018; Mauritsen et al., 2019), consisting of the spectral atmosphere general circulation model ECHAM6.3 (Giorgetta et al., 2012), the land surface vegetation model JSBACH3.2 (Raddatz et al., 2007) and the primitive equation ocean model MPIOM1.6 (Marsland et al., 2003). Two different resolutions are used for the simulations. 1) For calculating the SMB over the last deglaciation, MPI-ESM is used in its coarse resolution (CR) setup, hereafter referred
to as MPI-ESM-CR. In this setup ECHAM6.3 has a T31 spectral resolution (approx. 3.75°) with 31 vertical hybrid $\sigma$-levels,





which resolve the atmosphere up to 0.01 hPa (Giorgetta et al., 2012), and MPIOM1.6 has a nominal resolution of $3°$, with two poles located over Greenland and Antarctica (Mikolajewicz et al., 2007b). The selected setup is a compromise between computational feasibility and model resolution. 2) We additionally use simulation with the low resolution version of MPI-ESM1.2, hereafter referred to as MPI-ESM-LR, where ECHAM6.3 has a T63 spectral (approx. $1.88°$) grid with 47 vertical

levels and MPIOM features a $1.5°$ nominal resolution. For model details see Mauritsen et al. (2019).

## 2.3 Transient Climate Simulations

**MPI-ESM Experimental Designs**

We performed different simulations with the MPI-ESM-CR setup: 1) simulations to investigate the SMB throughout the last deglaciation, starting 26 ka BP until present, and 2) simulations to evaluate the SMB for historical climate conditions (see

Table 2). For the deglaciation experiment, the model was started from a glacial steady state and integrated from 26 ka BP until the year 1950, with prescribed atmospheric greenhouse gases (Köhler et al., 2017) and insolation (Berger and Loutre, 1991). The ice sheets and surface topographies were prescribed from the GLAC-1D (Tarasov et al., 2012; Briggs et al., 2014) reconstructions (Kageyama et al., 2017, see standardized PMIP4 experiments). The first 5 ka of this simulation are considered as spin-up, and we focus our analysis on the last 21 ka of the simulation. Hereafter, this simulation is referred to as MPI-ESM-

CR deglaciation experiment. All forcing fields are updated every 10 years and initiate changes of the topography and glacier mask, as well as modifications of river pathways, the ocean bathymetry and the land-sea mask (Riddick et al., 2018; Meccia and Mikolajewicz, 2018). Anthropogenic forcing, such as land use, is turned off in this simulation. For calculating the SMB, relevant variables of the atmospheric component of MPI-ESM1.2 are written out hourly throughout the simulation. Using the obtained atmospheric fields within the EBM (see Section 2.1) results in 3-D SMB fields, which are then interpolated onto the

GLAC-1D topography and ice mask (Tarasov et al., 2012). Computing a 3-D SMB also allows us to calculate the equilibrium line altitude (ELA), the elevation at which the SMB equals zero. Above this altitude it is thermodynamically possible to accumulate snow throughout the year and form an ice sheet or glacier. The ELA is less sensitive to changes in the ice sheet mask and less dependent on the surface topography than the SMB; hence, it is a good proxy for climate changes affecting the ice sheets.

For the evaluation of the derived SMBs under historical climate conditions, we branched off the last millennium simulation at 950 a BP from the deglaciation experiment. Topography, land-sea and glacier masks, river pathways, and the ocean bathymetry are taken from the deglaciation experiment and kept constant at 950 a BP. Other forcing fields are adopted according to the PMIP3 standard protocol for the Last Millennium simulations (Schmidt et al., 2012) and updated every year. The forcing fields for the years 1850 to 2010 are taken from the CMIP6 simulations (see Mauritsen et al., 2019). For the years beyond 2010, the

forcing fields in the desired resolution were not available at the time of the analysis. Overall, applied forcing allows for a more realistic treatment of atmospheric processes associated with changes in e.g., ozone, aerosols, $CO_2$ concentrations, and land use, and it accounts for their climatic impacts for present day climate conditions. Specifically, we apply time-varying greenhouse gases ($CO_2$, $N_2O$, $CH_4$), volcanic forcing, ozone, tropospheric aerosols and land-cover changes (see Junclaus et al., 2010;





Mauritsen et al., 2019). For the evaluation only the years 1980–2010 are used, which allows for a long enough adjustment of
the model to changes in the forcing. We hereafter refer to this simulation as the MPI-ESM-CR historical experiment. Note, that
changes in the topography due to ice sheets are small between 950 a BP and 2010. Hence, we expect only a minor impact on
the obtained SMBs.

Additionally, we performed a deglacial, last millennium, and historical simulation where ice sheets and topographies are
prescribed from ICE-6G (Peltier et al., 2015; Argus et al., 2007) reconstructions, an alternative reconstruction often used as
boundary forcing in deglacial simulations (Kageyama et al., 2017). Results from these simulations, hereafter referred to as
MPI-ESM-CR$_{Ice6G}$ experiments, are shown in Appendix A1 and emphasize differences in the SMB and relevant fields due
to different boundary conditions. While the SMB response to the climate forcing in these simulations is qualitatively similar
to the MPI-ESM-CR simulations forced with GLAC-1D reconstructions, differences in the freshwater runoff between the
reconstructions lead to a different climate response in the model simulations.

For a thorough evaluation of the EBM and in order to investigate the effect of model resolution on the historical SMB, we
additionally calculate the 3-D SMB fields from a CMIP6 (Wieners et al. , 2019) historical simulation with the MPI-ESM-LR
setup (see Section 2.2) (Mauritsen et al., 2019; Wieners et al. , 2019). The simulation allows to further evaluate the EBM
and to investigate differences in SMBs in regards to the spatial model resolution as well as differences due to the underlying
topographies. This simulation is hereafter referred to as MPI-ESM-LR historical experiment.

The 3-D fields derived for all historical control simulations are 3-dimensionally interpolated onto the ISMIP6 topography
and masked with the ISMIP6 ice mask (see Section 2.3 Nowicki et al., 2016; Fettweis et al., 2020).

**ERA-Interim Reanalysis**

To evaluate the EBM with respect to the atmospheric forcing data and its resolution, we additionally force the EBM with
ERA-Interim reanalysis data from the European Center for Medium-Range Weather Forecasts (ECMWF, Dee et al., 2011). For
comparison, we also interpolate the ERA-Interim derived 3-D SMB fields onto the ISMIP6 topography and masked with the
ISMIP6 ice mask (see Section 2.3). ERA-Interim is available as 6-hourly data on a $0.75°\times0.75°$ horizontal resolution. ERA-
Interim assimilates a great fraction of in-situ and remote sensing observations, making it one of the best reanalysis products
available (Cox et al., 2012; Zygmuntowska et al., 2012). However, as reanalysis data sets are model products, they exhibit
biases specifically for variables associated with small-scale processes, e.g., clouds, and areas where in-situ observations are
sparse (Stengel et al., 2018). These biases are not unique to the ERA-Interim but can be found in other reanalyses (Miller et
al., 2018). Relevant biases for this study are discussed in Section 3.

**The Regional Model MAR**

For evaluation, we compare the obtained SMB data sets from the MPI-ESM historical experiments to SMBs derived with
the regional climate model MAR (Modèle Atmosphérique Régional, version 3.9.6). For a detailed description of MAR and
its setup, see Fettweis et al. (2017, 2020). The MAR simulation used in this study was run on a 15 km horizontal resolution





with ERA-Interim boundary forcing (Dee et al., 2011) and interpolated onto the ISMIP topography (Nowicki et al., 2016), as described in Fettweis et al. (2020).

## 3 The Greenland Surface Mass Balance Under Historical Climate Conditions

For the Greenland ice sheet, a thorough evaluation of the accumulation and surface energy budget that determines surface
melt is conducted under historical climate conditions (1980–2010). Variables derived from the EBM simulations forced with output from ERA-Interim and the historical MPI-ESM1.2 simulations are compared to SMBs from MAR. SMB, accumulation and melt data sets are presented on the same ISMIP topography (see Section 2.3). All variables obtained using the EBM are hereafter referred to as $EBM_{MPI-ESM-CR}$, $EBM_{MPI-ESM-LR}$, and $EBM_{ERAI}$ for the EBM simulations forced with the historical simulations of both MPI-ESM-CR and MPI-ESM-LR, and ERA-Interim reanalysis. The annual mean SMB
averaged over 1980–2010 and the Greenland integrated value are shown in Figure 1 and Table 3 for each of the simulations. The corresponding plots for the MPI-ESM-CR$_{Ice6G}$ simulation are shown in Fig. A1.

The SMBs from $EBM_{ERAI}$, $EBM_{MPI-ESM-LR}$, and $EBM_{MPI-ESM-CR}$ show good agreement with SMBs from MAR for the historical period. The largest mass loss occurs along the low elevation areas close to the coasts, with maxima in the west and southwest of Greenland. The largest mass gain is evident in the higher elevation areas in the west and southeast of
Greenland. For all simulations, the mass changes over northern central Greenland are small, due to low precipitation at high elevations (see Fig. 2). Also, the gradients between areas of most pronounced mass loss and gain are qualitatively similar in all simulations. Differences in the SMB fields are largest along the coasts in the southeast and west of the Greenland ice sheet. These differences are likely a result of the forcing data, model resolution (about $3.75°$ – approx. 400 km at the Equator – for MPI-ESM-CR, $1.88°$ – approx. 210 km – for MPI-ESM-LR, $0.75°$ – approx. 80 km – for ERA-Interim, and 15 km for
MAR) and underlying topographies (Fig. 2). Differences between MAR and $EBM_{ERAI}$ are generally smaller than differences between MAR and $EBM_{MPI-ESM-CR}$ or $EBM_{MPI-ESM-LR}$ because ERA-Interim data were used as boundary forcing for the MAR simulation. Comparing SMBs derived from $EBM_{MPI-ESM-CR}$ and $EBM_{MPI-ESM-LR}$ shows that specifically the SMB differences in the North of the Greenland ice sheet, associated with more melt in $EBM_{MPI-ESM-CR}$ along the coasts and enhanced accumulation in the center of the ice sheet as compared to $EBM_{MPI-ESM-LR}$, are likely a consequence of the model
resolution. In the following we investigate the components that determine the SMB individually to understand the mentioned differences between the simulations better.

**Accumulation**

Accumulation patterns in MAR and the three EBM simulations are similar. However, they show some differences in the low elevation areas in the southeast of the ice sheet and the northern plateau (Fig. 1). Integrated over the ice sheet, $EBM_{ERAI}$,
$EBM_{MPI-ESM-CR}$, and $EBM_{MPI-ESM-LR}$ simulate lower accumulation than MAR as well as RACMO (Table 3). This difference is associated with less snowfall and more rainfall in ERA-Interim and the two MPI-ESM simulations than in the regional models, specifically in the low elevation areas along the coastal areas of Greenland (Fig. 2). The underestimation of



ERA-Interim's snowfall that extends into the high elevation areas of the ice sheet is likely associated with an unrealistic representation of clouds and a low cloud bias as well as shortcomings in modeling seasonal changes in surface temperatures (see
Section Melt; Miller et al., 2018). In the higher elevation areas of most of the central parts of Greenland MPI-ESM-CR and MPI-ESM-LR overestimate snowfall, which is tightly linked to topographic differences underlying the models (Fig. 2). Areas that are lower in MPI-ESM-CR and MPI-ESM-LR than MAR, mainly due to the spectral smoothing in MPI-ESM, generally show more snowfall in the MPI-ESM simulations (Fig. 2). Comparing the accumulation derived from $EBM_{MPI-ESM-CR}$ with $EBM_{MPI-ESM-LR}$ shows that accumulation patterns in the southeast of the ice sheet are more confined towards the east coast
but $EBM_{MPI-ESM-LR}$ still features a significant underestimation of accumulation in the low elevation areas. Hence, the even higher resolution of MPI-ESM-LR is not sufficient to represent the regionally confined processes that determine the accumulation in these regions. The overestimation of accumulation in the North of the ice sheet is reduced in $EBM_{MPI-ESM-LR}$ as compared to $EBM_{MPI-ESM-CR}$, which is likely associated with a better representation of the topographic gradients in the MPI-ESM-LR version of the model and an associated shift in precipitation patterns reducing precipitation at higher elevation.
Note, that the EBM calculates snowfall as precipitation at temperatures below 0°C and partly compensates for these differences in snowfall and rainfall, specifically along the coastal areas in the west and southeast of the ice sheet (not shown). The seasonal differences are larger. In summer, ERA-Interim simulates less snowfall and more rainfall but shows slightly less total precipitation than MAR, which impacts the melt patterns (not shown).

**Melt**

$EBM_{MPI-ESM-CR}$ shows significantly more surface melt along the western margins of the ice sheet than MAR (Fig. 1). These areas are topographically higher in MPI-ESM-CR than in MAR (Fig. 2). One problem of downscaling melt in these regions is that temperatures are always at the melting point during melting. By projecting the temperatures onto lower elevations, the height corrected temperatures depart significantly from the melting point towards higher temperatures. Hence, the vertical downscaling from higher elevations to low elevations overestimates melting. In contrast, the area in the south that is
significantly higher than the ISMIP topography shows less melt. It indicates that most of the differences are closely related to differences in the topography. Comparisons with $EBM_{MPI-ESM-LR}$, which shows less melt in the north and west of the ice sheet as compared to $EBM_{MPI-ESM-CR}$ (Fig. 1 and 2), confirm that differences in the melt patterns are linked to the underlying topographies of the model versions. MPI-ESM-LR is slightly higher than MPI-ESM-CR and thereby closer to MAR on the northern and western flanks of the ice sheet, hence $EBM_{MPI-ESM-LR}$ shows less melt than $EBM_{MPI-ESM-CR}$ in these
areas. $EBM_{ERAI}$ shows less melt in the southern and western parts of the ice sheets than MAR. These low melt rates are partly a result of the model tuning towards a similar integrated Greenland SMB value (see Table 3).

Heat fluxes towards the surface control predominately surface temperatures and melting. Miller et al. (2018), who compared surface energy fluxes over Greenland from different reanalyses with surface observations, found that ERA-Interim largely underestimates downward longwave and shortwave radiation, likely associated with an unrealistic representation of cloud
optical properties. Low surface albedos in ERA-Interim and an associated underestimation of outgoing shortwave radiation partially compensate for the downward longwave radiation deficit. Further, seasonal biases in the latent heat fluxes dampen





the seasonal changes in surface temperatures. Such biases are not unique to the ERA-Interim but can also be found for other reanalyses (see Miller et al., 2018, for details) and models, such as MAR (Fettweis et al., 2017). We find similar biases in the $EBM_{MPI-ESM-LR}$ simulation, which are likely associated with the simulated cloud cover.

## 4 SMB and ELA Changes Throughout the Last Deglaciation

The evaluation shows that most of the discrepancies between the SMBs of MAR and $EBM_{MPI-ESM-CR}$ are the result of differences in the model resolution and the underlying topography. Major differences are the increased melt on the western flank of Greenland and along the coastal areas as well as the overestimation in accumulation in the southern part of Greenland, which are largely a result of the model resolution as shown by comparisons with $EBM_{MPI-ESM-LR}$. Given these model limitations, the SMB is modeled well in comparison to MAR (or other regional models)(see also Fettweis et al., 2020) with the advantage of reduced computational costs that allow for a thorough investigation of the SMB for long-term climate simulations.

In the following we present the climate of the deglaciation experiment with MPI-ESM-CR based on Glac1D boundary conditions. The scope of this paper is not to evaluate the model simulation nor the forcing data sets used here, e.g., the ice volume changes as prescribed by the Glac1D or Ice6G reconstructions. Rather, we aim at exploring SMB and ELA changes under a transient climate forcing in order to understand the mechanisms behind their variability on glacial time scales. We limit the analysis to the Northern Hemispheric ice sheets only, with a specific focus on Greenland.

### 4.1 Greenland

As the SMB is highly dependent on the prescribed ice sheet geometry, it is challenging to interpret SMB changes for ice sheets that undergo substantial geometry changes throughout the deglaciation. As all ice sheets except Greenland collapse entirely, we investigate the SMB evolution mainly for Greenland, where changes in the geometry were relatively small (Fig. 3, gray line in the top panel). Values for the SMB, ELA, accumulation, and melt integrated over Greenland are shown in Fig. 3. The SMB and ELA for six time slices of the deglaciation are shown in Fig 4.

Cold Northern Hemispheric temperatures during the LGM (approx. 21 to 19 ka BP) are associated with a positive Greenland-wide integrated SMB of about 380 GT $a^{-1}$. This SMB is dominated by accumulation while melt is close to zero (Fig. 3 and 4). This result is consistent with the Greenland ice sheet being close to its maximum extent during this period (Clark et al., 2009). Due to the increase in temperatures, following an increase in Northern Hemispheric summer insolation by approximately 7% of the LGM value, and a simultaneous increase in the global $CO_2$ concentrations from 187 ppmv at 19 ka to 228 ppmv at 15 ka BP, both accumulation and melt increase. The total accumulation over Greenland increases from about 420 Gt $a^{-1}$, at 19 ka BP, to about 670 GT $a^{-1}$, at 15 ka BP (more than 35%). The largest accumulation increase is evident over the south-western part of the ice sheets, which is associated with more precipitation (Fig. 5). Intriguingly, the increase in precipitation is not a uniform signal for the entire Northern Hemisphere but shows regional patterns, such as a decrease over parts of the North Atlantic and south of the Laurentide ice sheet edge. These patterns indicate that precipitation changes are not entirely thermodynamically driven (the atmosphere being able to hold more water with increasing temperatures) but points towards



changes in the atmospheric dynamics. Melt increases from about 0 to 25 Gt a$^{-1}$ between the LGM and 15 ka BP. The growing
melt is small and limited to the low elevation areas along the coast of Greenland. This growth is a consequence of increasing
summer temperatures that exceed the freezing point in these areas and lead to enhanced melt during summer. In the other
areas over Greenland, temperatures are, despite warmer summers, still too cold to trigger melt. As the increase in accumulation
dominates enhanced melting, the SMB time series increases until about 15 ka BP (Fig. 3 and 4). Interestingly the ELA increases
despite an SMB increase. Per definition, the ELA depends directly on shifts in areas of net melt and accumulation. Hence, it
closely follows the increase in the ablation area. From the LGM to 15 ka BP the area of net ablation increases from 0 to about
58,400 km$^2$.

A simultaneous increase in SMB and ELA seems to be counter-intuitive at first, given that in a present-day climate, a
decrease in SMB over Greenland is associated with an increase in the ELA and vice versa (e.g., Le clec'h et al., 2017). As the
climate warms, the area of net ablation expands while the area of net accumulation shrinks, which moves the ELA upward. As
melt is close to zero in the glacial climate, the SMB is dominated by the significant growth of accumulation due to warmer
atmospheric temperatures and the associated increase in precipitation. The dominance of the accumulation in controlling the
SMB explains the counter-intuitive behavior of the SMB and ELA in the glacial climate. Further, it points towards the fact that
changes in the ELA cannot be taken as a proxy for changes in the SMB.

At around 14.6 ka BP the SMB and ELA over Greenland decrease significantly for about 500 a, the SMB drops from about
630 to 380 Gt a$^{-1}$ and the ELA decreases from more than than 460 m to 120 m (Fig. 3). Regionally differences are even
larger (Fig. 6). These drastic changes are associated with a significant reduction in the AMOC, as a response to increased
inflow of freshwater from melting ice sheets into the global ocean as prescribed from the Glac1D ice sheet reconstructions.
The strong melt-water pulse leads to a near-shutdown of the thermohaline circulation and a significant cooling of the North
Atlantic and adjacent regions (Fig. 6). Although the largest cooling is occurring over the North Atlantic, the annual cooling
signal extents over large regions of the Northern Hemisphere, including the Arctic Ocean, the North Pacific and large parts of
Eurasia and North America (Fig. 6). Over Greenland, this cooling diminishes surface melt during summer, which is similar to
LGM conditions. Again, the largest response is evident over the low elevation areas along the southern coasts of Greenland
(see also Fig. 5 for similarities). Associated with the overall cooling is a decline in precipitation, which reduces accumulation
by more than 40% over the ice sheet. Although melt and accumulation again partly compensate, accumulation changes occur
over a much larger area and dominate changes in melt, so that the integrated SMB decreases for Greenland (Fig. 3).

After the recovery of the AMOC, at around 14 ka BP, the SMB declines and the ELA continues to move upward. Thereby
it follows the overall warming signal as a response to increasing insolation and atmospheric greenhouse gases (Fig. 3 and
4). The decline in the SMB is associated with an overcompensation of the accumulation by a significant increase in melting.
Thus, ELA and SMB are anti-correlated from about 14 ka BP onward and continue to increase and decrease, respectively.
Only at around 13.6 ka and 11.6 ka BP, the ELA and SMB decrease significantly again, due to a second and third weakening
of the AMOC. Similar to the first AMOC decline, the associated cooling of the North Atlantic and parts of Greenland leads
to a decrease in accumulation, melt, and the ELA (Fig. 7). The changes are regionally very similar to the first event (Fig. 6).





However, the Greenland integrated SMB shows a weaker signal in both cases than during the first freshwater event, as the changes in accumulation and melt partly compensate if integrated over the ice sheet (Fig. 3).

After the 11.6 ka BP AMOC event, the retreat of the Greenland ice sheet towards his present-day state continues, associated with a decrease in SMB and an increase in ELA. The minimum SMB (216 Gt a$^{-1}$) is reached at 8.7 ka BP, and the maximum ELA (1556 m) occurs at 9.3 ka BP (Fig. 3 and 4). Due to the continuing deglaciation, Greenland experiences its largest ice volume and extent changes between 10.8 ka and 9.1 ka BP (the ice sheet geometry influences SMB values during this period). At around 11.1 ka BP, the Northern Hemispheric summer insolation reaches its maximum and decreases continuously

thereafter until the present, while the $CO_2$ concentration remains rather constant between 11.1 ka and 6 ka BP and slightly increases thereafter. Consequently, the ELA decreases, and the SMB begins to recover continuously after about 8.7 ka BP. Note, that a series of smaller AMOC weakening events is evident at around 10.1 ka, 8.4 ka, and 7.1 ka BP, but their climate impact on the ELA and SMB does not manifest significantly in the time series for Greenland (Fig. 3). The ELA decrease and SMB increase continue until 200 a BP despite a slight increase in the $CO_2$ concentration. These continuing changes suggest

that the decreasing summer insolation drives SMB and ELA changes between 9 ka and 200 a BP. It is not before 100 a BP that the ELA and SMB closely follow the $CO_2$ signal again. The sharp drop in the SMB and the uplift of the ELA for the last 100 years of the simulation is similar to the warm period observed in the coastal temperatures of Greenland in the 1930s (Chylek et al., 2006). At the end of the simulation the ELA lies at about 1150 m and the SMB reaches values of 550 Gt a$^{-1}$. These values are similar to values observed during the 21st-century (see Fig. 4, Section 3 and Table 3; Box, 2013).

The SMB and ELA derived from the MPI-ESM-CR simulation with the prescribed Ice6G ice sheets are qualitatively similar to the presented results based on the Glac1D ice sheet reconstructions (see Fig. A2 to A6). The overall trends of both variables are similar, but the timing of the weakening of the AMOC as well as the magnitude differ. These deviations are due to a different timing, magnitude and location of melt water release between the Glac1D and Ice6G reconstructions.

### 4.2    Impact on the Eurasian and North American ELA

As discussed earlier, the North American (Laurentide and Cordilleran) and Eurasian (Fennoscandian, British Isles, and Barents-Kara Sea) ice sheets experience substantial changes in their glacial extent throughout the last deglaciation (Fig. 4). Reconstructions suggest a steady retreat of the Eurasian ice sheets starting from the LGM until about 9.7 ka BP, when the last ice vanished from the continent. The decline is highly discontinuous and shows an acceleration at around 18.5 ka BP. Similarly, the extent of the North American ice sheet started to decrease shortly after the LGM and continued until about 6 ka BP, with only little

ice left to the present day. As the SMB is highly dependent on the ice sheet geometry, it is difficult to fully interpret SMB changes for the North American and Eurasian ice sheets. Hence, we only briefly point out the similarities between the climate responses of the ELAs over the North American, Eurasian, and Greenland ice sheets (see Fig. 5, 6 and 7) while keeping in mind that the interpretation is limited due to the extensive changes in the ice sheet geometries throughout the deglaciation. To account for the massive ice sheet changes in the time series, we split the ice sheets into different sub-regions and investigate

their dependence on the ice sheet geometry. Fig. 8 shows the ELA time series for Eurasia, subdivided into a southern (<60°N),





central (60°-70°N) and northern (>70°N) ice sheet, and North America, split into a northern and southern Laurentide ice sheet (east of 120°W and north of 60°N and south of 60°N, respectively) and a Cordilleran ice sheet (west of 120°W).

At the LGM, the average ELAs over Eurasia and North America are significantly higher than the ELA over Greenland for the same period, except for the North Eastern Laurentide and Northern Eurasian ice sheets (see Fig. 4 and 8). Similar to the

Greenland ice sheet, the ELA increases continuously until about 15 ka BP for the Eurasian and North American ice sheets. At around 14.5 ka BP the southern and central Eurasian ice sheets both show a slight decrease in the ELA due to the AMOC slowdown (see Section 4.1); other AMOC events at around 19.6 ka, 18.2 ka, and 15.2 ka BP also result in a decrease in ELA for the southern and central Eurasian ice sheet. However, the signal in the ELA is relatively weak and regionally confined compared to the response over Greenland. Around 14.5 ka the Eurasian ice sheet exhibits decreased ELAs on its western

boundaries and the Laurentide ice sheet on its eastern boundaries since both melt and accumulation decrease in response to reduced North Atlantic temperatures (Fig. 6). This result suggests that the AMOC slowdown at 14.5 ka affected all ice sheets, at least regionally. After this event, the ELA continues to increase for nearly all ice sheets, following the overall warming signal. However, specifically for the southern ice sheets over Eurasia and North America the changes in the ice sheet extent have a large effect on the ELA. The signal in the ELA due to the changes in the ice sheet mask exceeds the signal due to

the natural climate variability of the ELA at around 20.6 ka BP for the southern Eurasian ice sheet, about 18.9 ka BP for the northern Eurasian and southeast Laurentide, about 15.3 ka for the northern and central Eurasian and 13.3 ka for the Cordilleran ice sheet (Fig. 8). Hence, the second and third AMOC slowdowns are only poorly reflected in the ELA time series, although similar regional responses are evident in Fig. 7. By 9.7 ka BP Eurasia is completely ice free, while North America has only small ice sheets left (see Fig. 4).

**5   Discussion**

Despite the relatively low resolution of the MPI-ESM-CR simulation, used as forcing for the EBM, the obtained SMBs for historical climate conditions show good agreement with SMBs derived from regional climate modeling (see Table 3). A comparison with SMBs derived from a simulation with the MPI-ESM-LR setup and ERA-Interim reanalysis reveal that discrepancies between the SMBs derived from MPI-ESM-CR and MAR are a result of the coarse resolution of the model (e.g., the

extensive melt in the North of the Greenland ice sheet) and due to the quality of the forcing itself (e.g., precipitation patterns). In contrast to ERA-Interim, MPI-ESM evolves freely and does not assimilate any surface observations; hence, differences are to be expected. Further differences are related to underlying topographies of the native models as well as the fact that most fluxes within the EBM are parameterized, as they are not directly available from the model simulations in the required temporal resolution. For example, several studies have pointed towards the specific importance of the albedo parameterization for a

realistic simulation of the SMB here (e.g., van Angelen, 2012). Here we chose a parameterization that yields realistic SMBs for Greenland for the historical period (see Section 3). Analyzing the MPI-ESM-CR deglaciation experiments for Greenland, we find that the SMB changes in the beginning of the deglaciation are associated with compensating effects of increasing accumulation and melt. The increase in accumulation dominates the increase in melt until about 14 ka BP, as significant melt



is not evident before. After 14 ka BP the SMB decreases, indicating that this time marks the onset of the deglaciation. The
ELA begins to increase significantly earlier, from about 19 ka BP, suggesting that the deglaciation of Greenland might have
been triggered earlier, likely by the insolation increase that set in before 21 ka BP. For Eurasia and North America, the ELA
increases continuously from 21 ka, supporting such a hypothesis. An onset of the deglaciation over Greenland at around 14 ka
BP is found in reconstructions and has been connected to the Bølling-Allerød warm period, which was associated with a strong
AMOC and warm North Atlantic temperatures (e.g., Clark et al., 2002; Weaver et al., 2003). In MPI-ESM-CR, we do not find
such warming but instead find an AMOC slowdown at 14.6 ka BP. This slowdown is caused by a meltwater pulse of more than
0.4 Sv, prescribed by the ice sheet reconstructions (meltwater pulse 1A). As this meltwater pulse is associated with ice volume
changes mostly in the North Atlantic/Arctic drainage basin in the ice sheet reconstructions, it is assumed that the meltwater
enters the North Atlantic and Arctic. This meltwater weakens the AMOC, which is associated with the strong cooling in the
North Atlantic realm, in our model. After the end of the prescribed melt water peak the AMOC recovers in the model with
some overshooting of the AMOC (peaking after 200–300 a). This result indicates that although our model does not simulate the
Bølling-Allerød warm period as represented in proxy data, the overall progression of the deglaciation is represented reasonably
well.

The AMOC slowdowns that occur throughout the last deglaciation are associated with significant changes of the ELA and
SMB, specifically over Greenland and regionally also for the Eurasian and North American ice sheets. They are associated with
cooling over the North Atlantic and a decrease in accumulation and melting in the adjacent regions. While the response in melt
is directly affected by temperature changes, accumulation not only directly depends on temperature, affecting the quantitative
distribution of snow- and rainfall, but also on other atmospheric properties, e.g., the capability of the atmosphere to hold
water, changes in the atmospheric circulation and convection (Trenberth, 2011). The differences in the ELA in response to the
AMOC slowdowns over the Laurentide and Eurasian ice sheets further away from the North Atlantic coasts are challenging to
interpret for the following reasons. Substantial changes in the ice sheet height and volume cause significant surface warming
and enhance melting specifically at the southern margins of the ice sheets (Fig. 6 or 7). Other contributing factors are more
difficult to separate due to the nature of the experimental design, such as the effect of sea-level changes on the ice sheet margins,
feedbacks in response to changes in the sea level, sea ice, and greenhouse gases. In addition, the missing ice sheet dynamics
might result in a modified ELA response due to differences in the ice sheet height and configuration (Cronin, 2010).

A comparison between ELAs and SMBs derived from MPI-ESM-CR simulations with different ice sheet reconstructions
(Glac1D and Ice6G) as boundary forcing reveals that the results are qualitatively similar (see Fig. A2 to A6). However, the
modeled changes in the AMOC are highly dependent on where the freshwater forcing is applied (e.g., Tarasov and Peltier, 2005;
Kleinen et al., 2009). The scope of this study is not to evaluate the forcing data sets used here, e.g., the ice volume changes
as prescribed by the reconstructions. All reconstructions pose unexplored uncertainties (Kageyama et al., 2017). However, the
comparison shows that the changes presented here are robust, specifically for changes in the ELA.
## 6  Conclusions

We have presented the first timeseries and analysis of SMB changes over the last deglaciation from a simulation with a comprehensive ESM. Derived SMBs for historical climate conditions compare well with SMBs from regional climate modeling. Discrepancies are related to the coarse model resolution, the accuracy of modeled climate patterns relevant for accumulation

and melt, as well as parameterizations used within the EBM. Given the computational advantages of the used coarse resolution version of MPI-ESM in connection with the EBM for long-term simulations, these limitations and differences are an acceptable compromise.

The deglaciation experiment suggests that the onset of the deglaciation was triggered by the insolation increase starting from 21 ka BP. The main climate features that characterize the last deglaciation are modelled reasonably well, although not

all climate features indicated from proxy data (e.g. the Bølling-Allerød warming) are simulated by our model. Differences in the SMB and ELA response throughout the deglaciation are dominated by compensating effects of accumulation and melt. Regional differences in the SMB and ELA response between Greenland, the Laurentide and Eurasian ice sheets are difficult to interpret, due to processes associated with changes in the glacial configuration and dynamical processes within the ice sheet, which are currently not included in our setup. The latter would require the incorporation of an interactive ice sheet model,

which is currently employed within the MPI-ESM setup in the scope of the project PalMod (Latif et al., 2016; Ziemen et al., 2019). Utilizing the SMB data set presented here as forcing for ice sheet model simulations will allow for an investigation of ice sheet dynamics.

*Code and data availability.*  MPI-ESM is available under the Software License Agreement version 2 after acceptance of a license (https://www.mpimet.mpg.de/en/science/models/license/). The 3-D SMB data set as well as ocean forcing required to conduct ice sheet model

experiments for the deglaciation experiments with different ice sheet reconstructions can be obtained from the DKRZ World Data Center for Climate (WDCC) at https://cera-www.dkrz.de/WDCC/ui/cerasearch/entry?acronym=DKRZ_LTA_989_ds00006 (Glac1D boundary conditions) and https://cera-www.dkrz.de/WDCC/ui/cerasearch/entry?acronym=DKRZ_LTA_989_ds00007 (Ice6G). Additional data sets are available from the corresponding author upon request.

## Appendix A:  Supplementary Figures

*Author contributions.*  MK, FZ and UM developed the idea for the manuscript. UM, CR and MK advanced the EBM model code and UM adapted MPI-ESM-CR for transient deglaciation simulations. UM performed the deglaciation and MK the millenniums and historical simulations with the MPI-ESM-CR setup. MK conducted the analysis and wrote the manuscript, with contribution to Section 2.1 by CR. CS, MK and FZ prepared the ice-sheet model forcing data. All authors commented and improved the manuscript.



*Competing interests.* The authors declare that they have no conflict of interest.

*Acknowledgements.* This project was supported by the German Federal Ministry of Education and Research (BMBF) as a Research for Sustainability initiative (FONA) through the PalMod project under grant no. 01LP1504C, 01LP1502A, 01LP1915 and 01LP1917. Christian Rodehacke has received funding from the European Research Council under the European Community's Seventh Framework Programme (FP7/2007-2013) grant agreement 226520 (COMBINE, EC IP) and (FP7/2007–2013)/ERC grant agreement 610055 as part of the Ice2Ice project. All simulations were performed at the German Climate Computing Center (DKRZ). We also acknowledge the ECMWF for granting
access to the ERA-Interim reanalysis. Furthermore, we like to thank Xavier Fettweis and Brice Noël for providing the MAR and RACMO data. We thank Stephan Lorenz for assisting with the setup and Matthew Toohey for providing the volcanic forcing for the last millennium simulations. We also thank Heinrich Widmann for assisting us to store our data at the World Data Center for Climate (WDCC) and the WDCC for hosting our dataset. Additionally, we are grateful to Thomas Kleinen for his helpful comments to improve this manuscript.



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





**Table 1.** List of constants used in the albedo scheme as part of the EBM. Please see Section 2.1 for further details.

| Description | Symbol | Value/Comment |
|---|---|---|
| Fresh fallen snow | $\alpha_{frsnow}$ | 0.92 |
| Firn | $\alpha_{firn}$ | 0.72 |
| Snow layer | $\alpha_{snow}(t)$ | between $\alpha_{frsnow}$ and $\alpha_{firn}$; Equation 1 |
| Ice albedo | $\alpha_{ice}$ | 0.70 |
| Final albedo of snow | $\alpha_{sfc}(t, d_{snow})$ | between $\alpha_{frsnow}$ and $\alpha_{ice}$; Equation 2 |
| Time scale of snow aging | $\hat{\tau}_a$ | 1/30 day$^{-1}$ |
| Time scale for aging of refrozen snow and ice | $\hat{\tau_a r}$ | 1/45 day$^{-1}$ |
| Depth scale of snow layer thickness | $\hat{d}$ | 0.0024 $m^{-1}$; Equation 2 |
| Melted snow/ice | $\alpha_{snow-/icemelt}$ | 0.55 |
| Refrozen snow | $\alpha_{snowrefrz}$ | 0.67 |
| Refrozen ice | $\alpha_{icerefrz}$ | 0.55 |
| Background albedo below snow layer | $\alpha_{bg}$ | for snow aging in Equation 1; defined in Equation 3, where $\alpha_{bg} \leq \alpha_{ice}$ |
| Density dependence of background albedo parameter, factor | $q_1$ | $-4\,10^{-4}$ m$^3$ kg$^{-1}$ |
| Density dependence of background albedo parameter, offset 1 | $q_2$ | 0.95 |
| Background albedo below refrozen layer | $\alpha_{bg}$ | $\alpha_{bg} = \alpha_{ice}$ for snow aging, Equation 1 |





**Table 2.** Simulations performed, as described in Section 2.3. For the deglaciation experiments the topography and ice sheets are taken from reconstructions and change throughout the simulation. For the historical and last millennium simulations those fields remain constant through time.

| Period | Experiment Name | Topography/Ice Sheet Mask | Spin-up |
|---|---|---|---|
| Deglaciation | MPI-ESM-CR | Glac1D (transient) | 26 kas BP steady state |
| (26k–0 ka BP) | MPI-ESM-CR$_{Ice6G}$ | Ice6G (transient) | 26 kas BP steady state |
| Historical | MPI-ESM-CR | Glac1D (950 a BP) | Last millennium simulation |
| (1850–2010) | MPI-ESM-CR$_{Ice6G}$ | Ice6G (950 a BP) | Last millennium simulation |
| | MPI-ESM-LR* | CMIP6 | Preindustrial steady state |

* This simulation was taken from CMIP6 (see Section 2.3).





**Table 3.** Average and standard deviation of the annual SMB simulated with MAR and the EBM forced with ERA-Interim and a historical MPI-ESM simulation for the years 1980–2010. The annual SMB values are interpolated onto the ISMIP ice sheet topography (see Section 2.3), except for RACMO$^*$. The standard deviation reflects the inter-annual variability. Units are in GT a$^{-1}$. Accumulation is calculated as residual of the SMB and melt.

| Model | SMB | Melt | Accumulation |
|---|---|---|---|
| MAR | 340±122 | -548 | 888 |
| RACMO$^*$ | 367±108 | -540 | 907 |
| EBM$_{ERAI}$ | 344±140 | -288 | 632 |
| EBM$_{MPI-ESM-LR}$ | 351±83 | -311 | 663 |
| EBM$_{MPI-ESM-CR \ (Glac1D)}$ | 275±107 | -495 | 770 |
| EBM$_{MPI-ESM-CR \ (Ice6G)}$ | 307±71 | -440 | 747 |

$^*$ RACMO data are provided on a slightly different topography with 1 km horizontal resolution (see Noël et al., 2019, for details); the impact of the underlying topography on the SMB values is expected to be small.



**Figure 1.** SMB (top), accumulation (middle) and melt (bottom) from MAR (left) and EBM simulations forced with ERA-Interim (center, left), MPI-ESM-LR (center, right) and MPI-ESM-CR (right) for historical climate conditions. The values are averaged over 1980–2010. All variables are interpolated on the ISMIP6 topography and shown only for glaciated points. Note that accumulation is obtained as residual of SMB minus melt, as MAR does not provide accumulation as direct output variable. Hence, the contour where accumulation transitions from positive values to zero represents the equilibrium altitude line (ELA). Black contours mark surface elevation at 1000, 2000 and 3000 m.





**Figure 2.** Total Precipitation (top) as simulated by MAR (left), ERA-Interim (center, left), MPI-ESM-LR (center, right) and MPI-ESM-CR (right) for 1980–2010. Snowfall (top, middle) and rainfall (bottom, middle) in MAR (left) and differences between ERA-Interim (center, left), MPI-ESM-LR (center, right) and MPI-ESM-CR (right) and MAR. Topography (bottom) from ISMIP6 (left) and the differences in topography between ISMIP6 and ERA-Interim (center, left), MPI-ESM-CR (center, right) and MPI-ESM-LR (right). Note, that the values are bi-linearly interpolated onto the ISMIP6 topography from the original model data, not the downscaled values. Black contours mark surface elevation at 1000, 2000 and 3000 m.



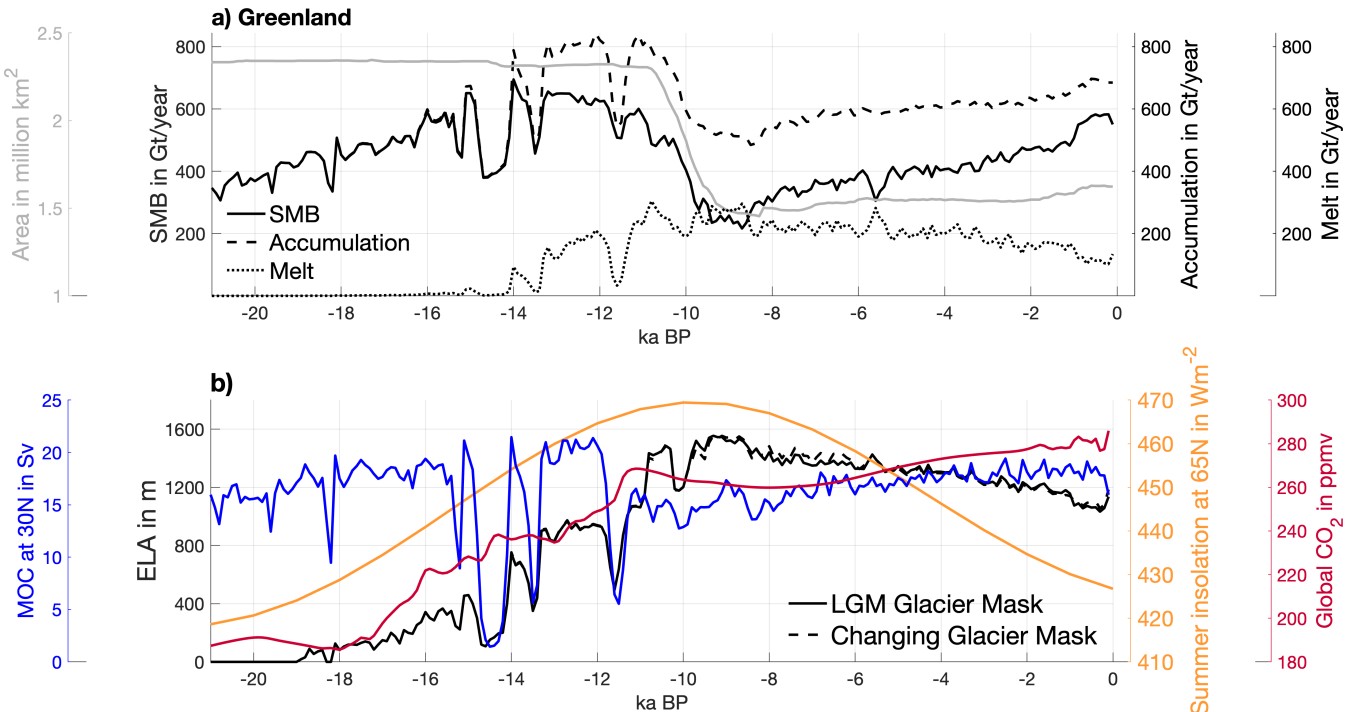

**Figure 3.** (a) Greenland SMB, accumulation, melt, ice sheet area and (b) Equilibrium Line Altitude (ELA), Meridional Overturning Circulation (MOC) for the EBM$_{\mathrm{MPI-ESM-CR}}$ experiment, together with summer insolation at 65°N and $CO_2$ concentration throughout the last deglaciation (21 ka to 0 ka BP). Here 0 ka BP refers to the year 1950. SMB, accumulation, melt and ELA (dashed) are integrated over the glacial mask of each individual 100-year time slice. Additionally, the ELA (solid) is integrated over the 21 ka BP ice sheet mask, in order to investigate differences due to the ice-sheet mask. The MOC is the overturning strength at 30.5°N at a depth of 1023 m as in Klockmann et al. (2016). The $CO_2$ concentration is taken from Köhler et al. (2017) and the summer insolation from Berger and Loutre (1991).

.




**Figure 4.** SMB and ELA for selected time slices (21 ka, 15 ka, 14 ka, 11 ka, 9 ka and 0 ka BP). Shown are 100-year means. The SMB is interpolated on the Glac1D topography for each individual time slice, the ELA is shown on the native MPI-ESM-CR model grid. Note that the different glacier masks are a result of the different resolution

.





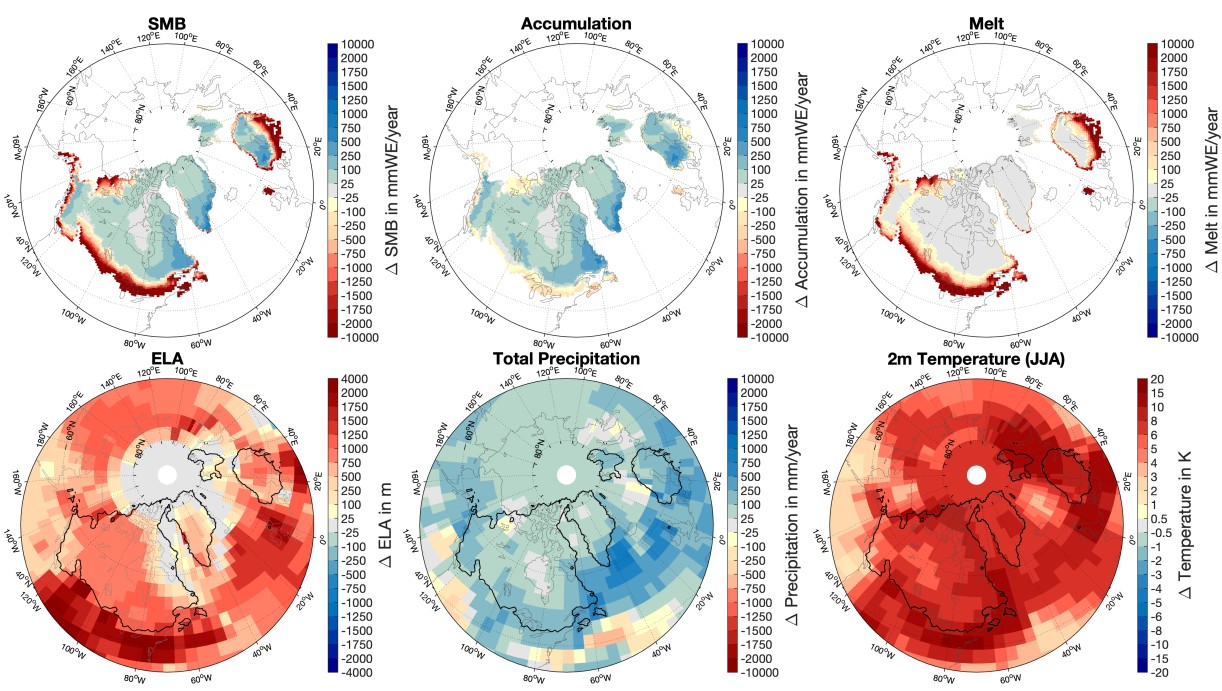

**Figure 5.** Differences of SMB, accumulation and melt, precipitation and 2-m temperatures as well as summer 2-m summer temperatures between 15 ka BP and the LGM. For the differences SMB, accumulation and melt are interpolated on the Glac1D topography of each individual time slice. The other variables are shown on the native MPI-ESM-CR model grid. Black contours in the lower panels indicate the ice-sheet mask at 15 ka BP.



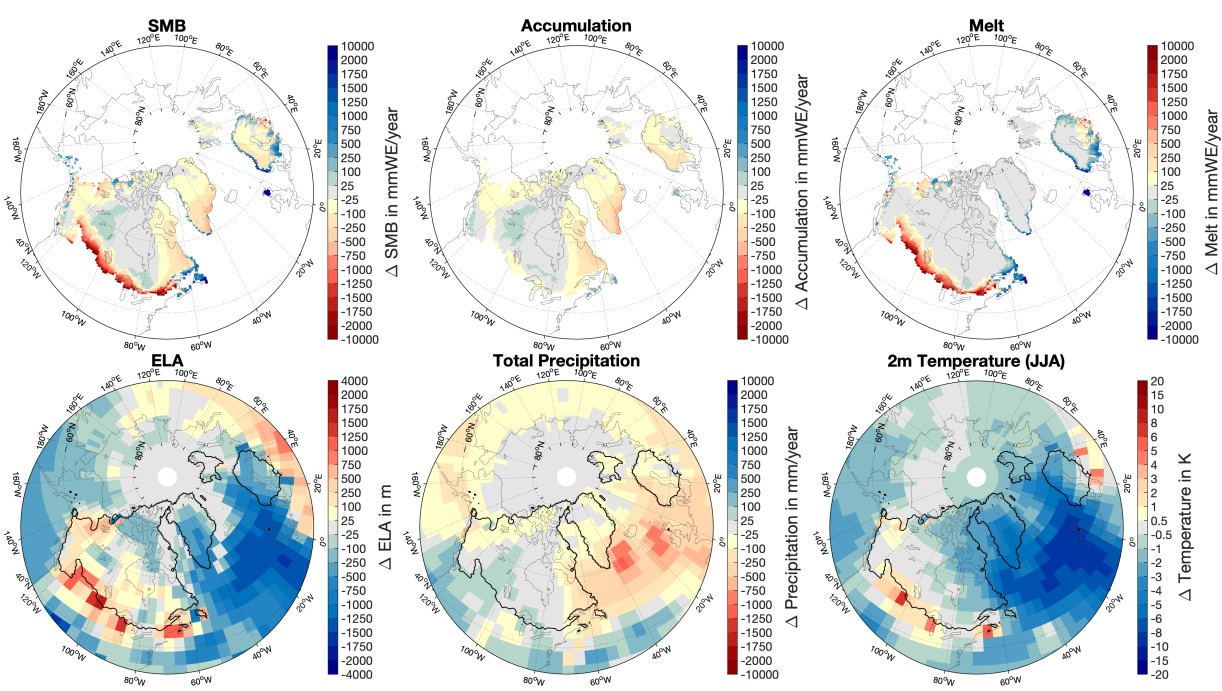

**Figure 6.** Similar to Fig. 5, but for differences between 14.6 ka and 15 ka BP.



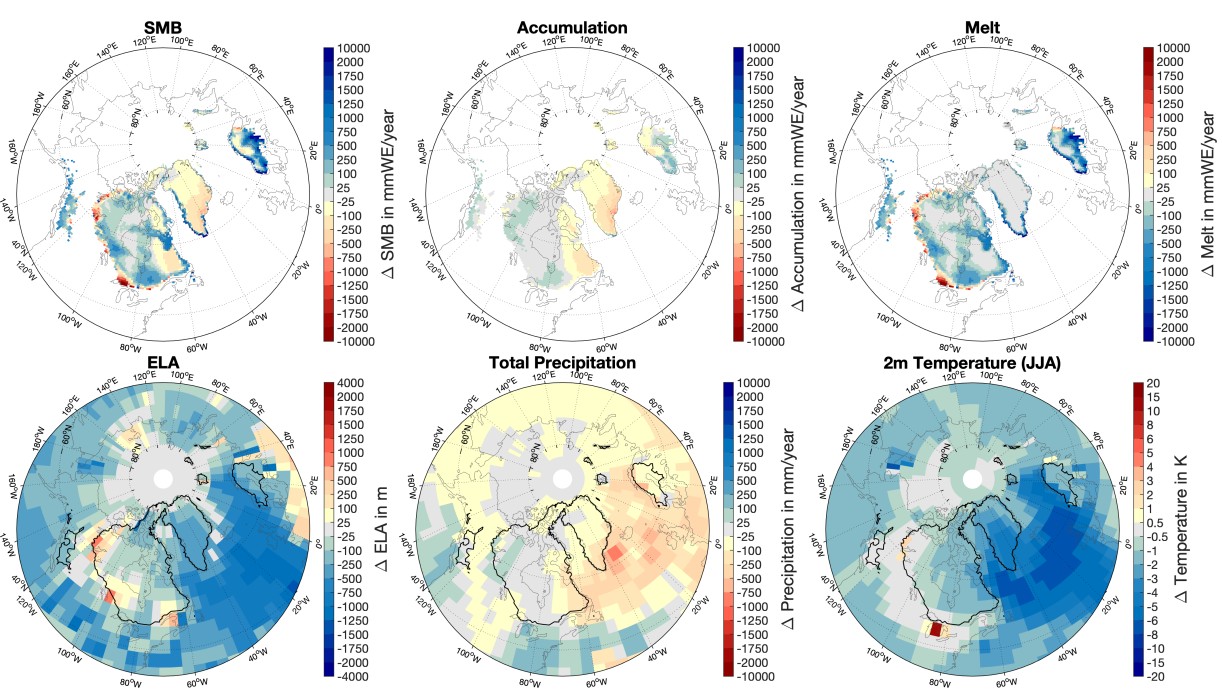

**Figure 7.** Similar to Fig. 5, but for differences between 11.6 ka and 12 ka BP.





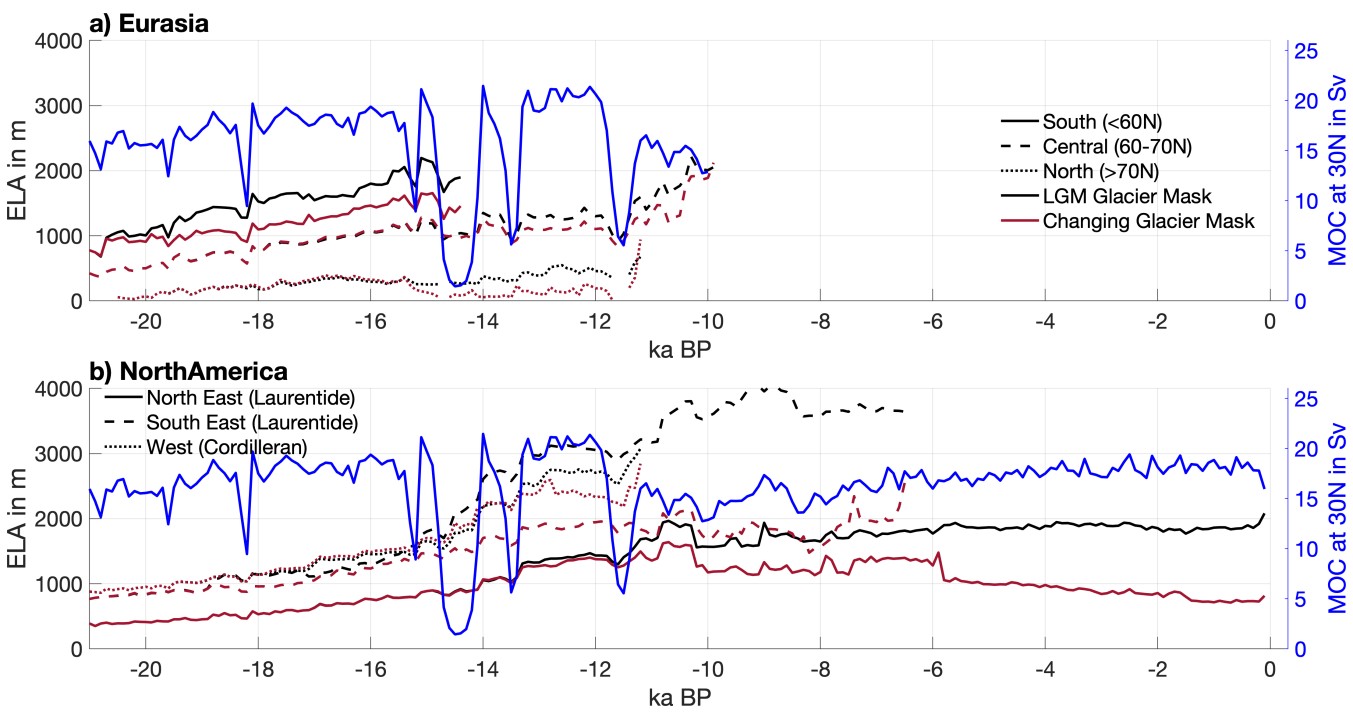

**Figure 8.** Similar to the bottom panel of Fig. 3 but for the sub-divided (a) Eurasian and (b) North American ice sheets. The Eurasian ice sheet is subdivided into a southern (<60°N), central (60°-70°N) and northern (>70°N) part. The North American ice sheet is split into the northern and southern Laurentide ice sheet (east of 120°W and north of 60°N and south of 60°N, respectively) and the Cordilleran ice sheet (west of 120°W). The ELA is integrated for each region over the 21 ka BP ice sheet mask (black) and over the ice sheet mask of each respective time slice (red).



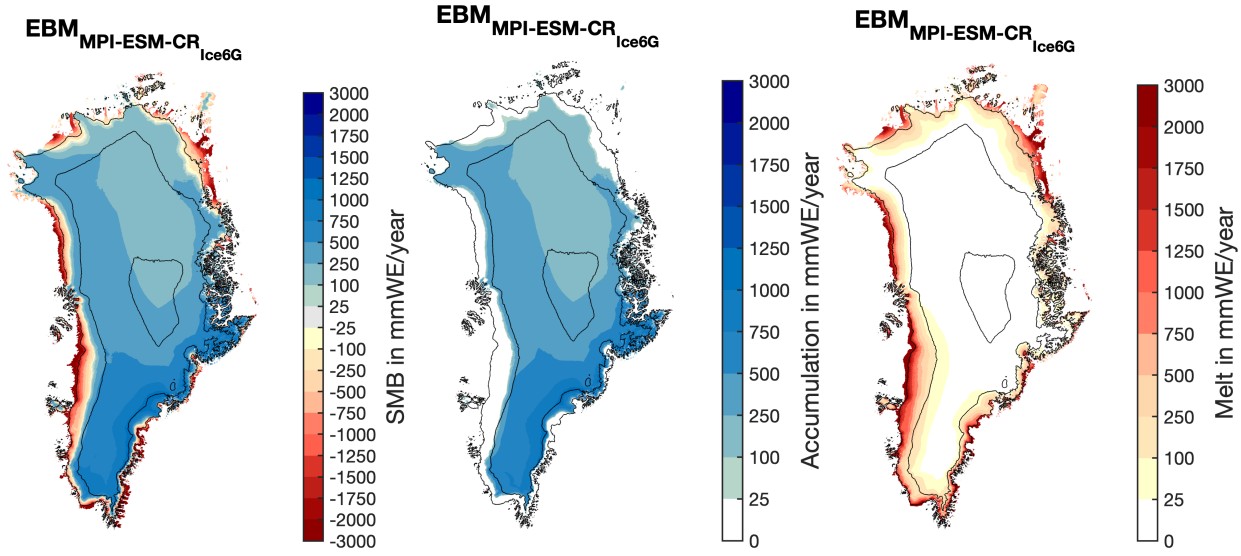

**Figure A1.** Similar to Fig. 1 but for the MPI-ESM-CR$_{Ice6G}$ simulation.





**Figure A2.** SMB and ELA for selected time slices (21 ka, 15 ka, 14 ka, 11 ka, 9 ka and 0 ka BP). Shown are 100-year means. The SMB is interpolated on the ICE-6G topography for each individual time slice, the ELA is shown on the native MPI-ESM-CR model grid. Note that the different glacier masks are a result of the different resolution

.





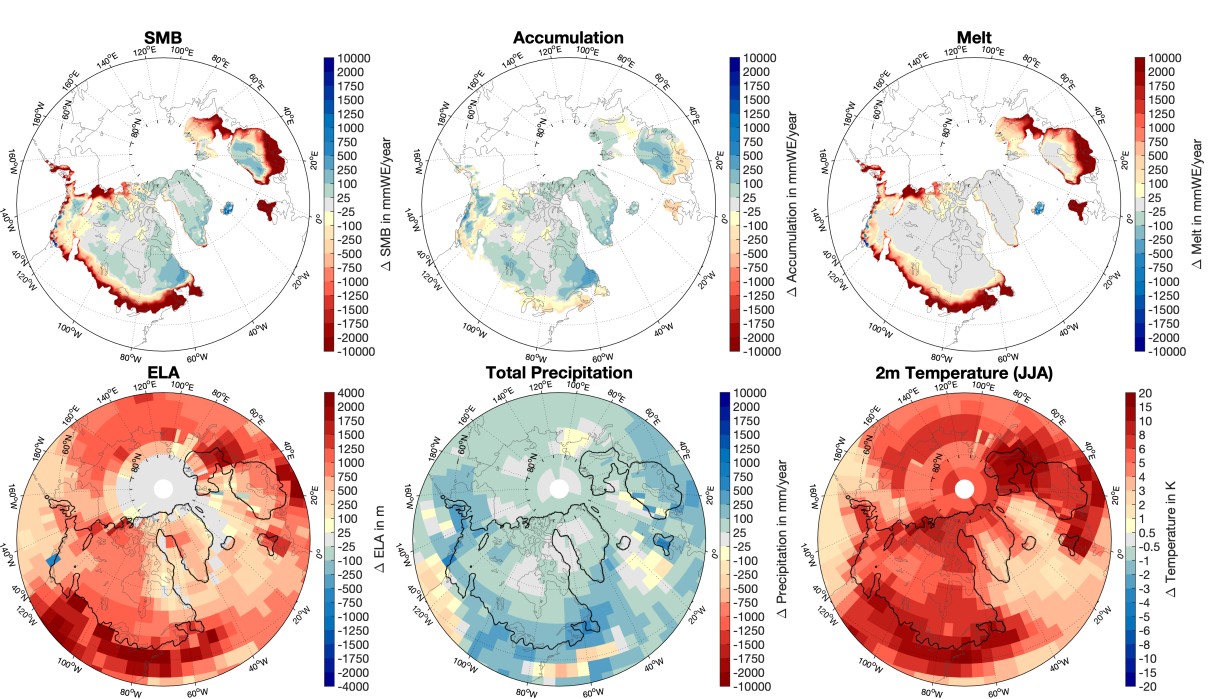

**Figure A3.** Similar to Fig. 5 but for differences between 15 ka and LGM from MPI-ESM-CR$_{\text{Ice6G}}$.
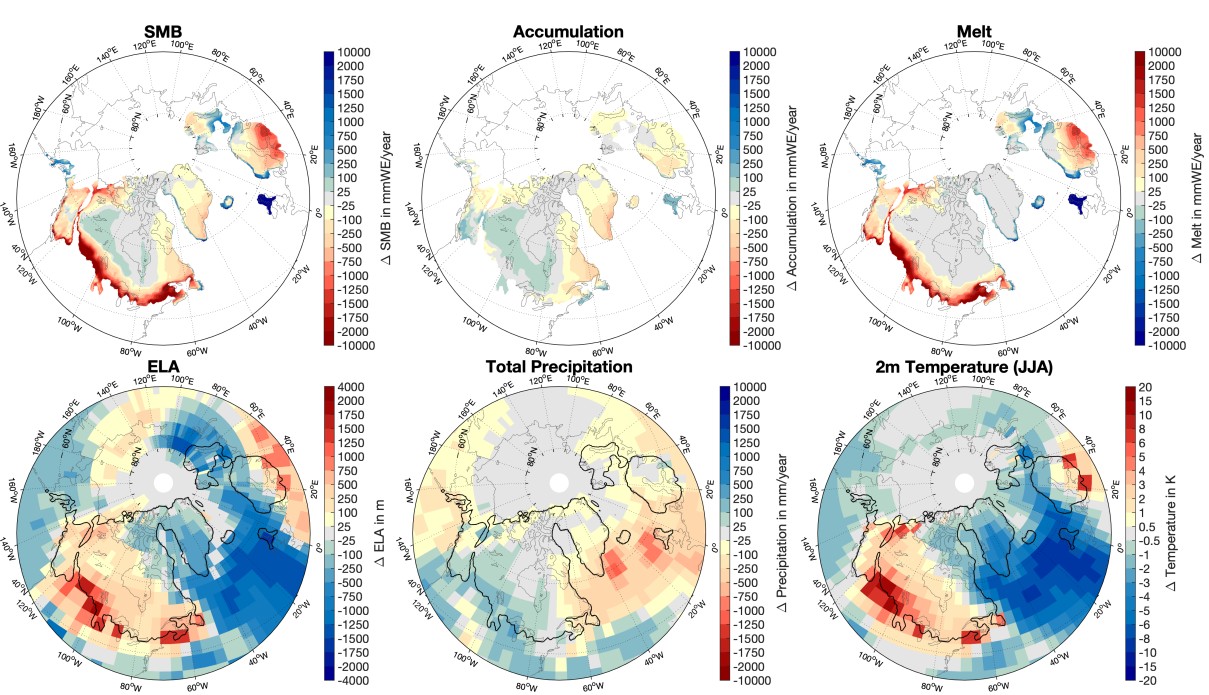

**Figure A4.** Similar to Fig. A3, but for differences between 14.4 ka and 15 ka BP from MPI-ESM-CR$_{Ice6G}$. Note, that a different time slice was chosen as compared to Fig. 6 due to a later AMOC slowdown in the MPI-ESM-CR simulation with Ice-6G boundary conditions.
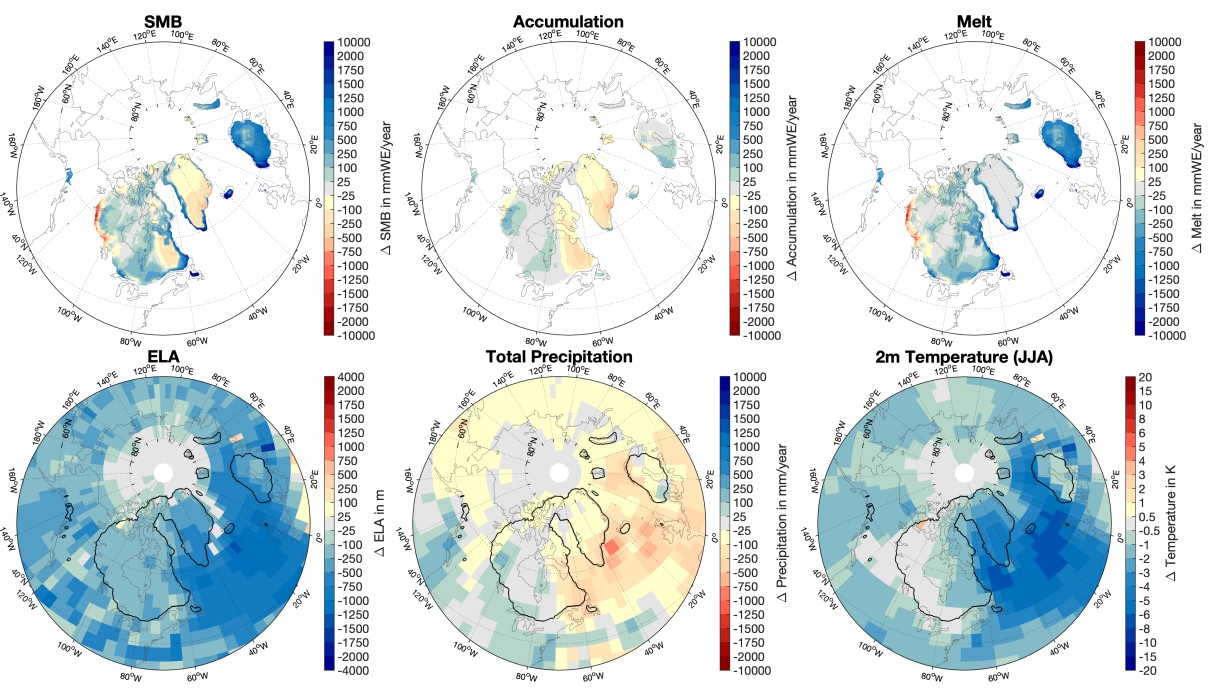

**Figure A5.** Similar to Fig. A3, but for differences between 11.2 ka and 12.2 ka BP from MPI-ESM-CR$_{\text{Ice6G}}$. Note, that a different time slice was chosen as compared to Fig. 7 due to different timings of the AMOC slowdown in MPI-ESM-CR with Ice-6G boundary conditions.







**Figure A6.** Similar to Fig. 3 and 8 but for EBM$_{MPI-ESM-CRIce6G}$ with prescribed Ice6G reconstructions.