# Peer review of "Analysis of the Surface Mass Balance for Deglacial Climate Simulations"

_The Cryosphere, 2020_

## Referee Comment (RC1) · Anonymous Referee #1 · 21 Sep 2020

This paper presents a downscaling energy balance surface/snow model, intended to derive more detailed surface mass balance fields for glacial regions in climate models. The downscaled surface mass balance may be analysed in its own right (as done here), or used as a boundary condition for driving an ice sheet model. The downscaling model is described, and its performance evaluated with reference to a regional climate model simulation of Greenland's recent past. The final section describes their calculation of surface mass balance fields for the northern hemisphere hemisphere ice sheets, derived offline from a climate simulation of the last deglaciation conducted with the coarse resolution version of the MPI-ESM. Running ice sheet models more closely with climate simulations is a growing field, and this model represents a very useful contribution to area. The paper is generally carefully written and covers the all the

necessary topics, although there are quite a few specific places I think it could do with a little clarification, as detailed below. More generally I think sharpening up the framing and motivation - especially for the paleoclimate section - would be valuable.

General comments

I did enjoy reading this, and it feels like I've spent a lot of time writing it up - apologies for the lateness - I found lots of small things to ask about, see below.

I felt there was one wider, general issue that hopefully would be easy enough to address - as I got to the paleoclimate section I didn't think that a solid motivation had really been given for /why/ a simulation of deglacial SMB was being done, and thus I wasn't really sure what to be looking for in how it's been described and analysed. Even by the end this isn't clear, as the Conclusions don't conclude anything that sounds obviously new about the glacial climate system or the model itself. In a couple of places it's said that the SMB fields will be made available to the ice sheet community for their use, but the first part of section 4 says that the simulation isn't going be evaluated, so I don't really buy this as sufficient motivation for the whole exercise - are these modelling groups going to want to use a new SMB product from a climate simulation that hasn't been evaluated for how it compares to evidence of what actually happened?

There is some interesting decomposition of how accumulation and melt factors balance differently in different periods and how the (perhaps non-obvious) relationship between total SMB and ELA changes too, but only for GrIS - is it different in different regions? - and no attempt is made at general statements about the global implications or testable hypotheses for underlying principles. Personally, I wanted to know more about how some of the details of the snow physics were being forced by the climate through the deglaciation - albedo, refreezing etc, rather than just the resulting SMB, but that's just one idea.

I think the authors should either decide on a clear main goal that's stated at the front of the paper and runs all the way through, or be more clear about the separate aims of

each part - either way that should make it easier to see what the paper as a whole is working towards, and clearer what should be in the final conclusions.

Detailed comments:

line 4: "deglacial climate" - quite specifically the /last/ deglaciation

line 5: "allows to resolve" isn't grammatically great - 'allows the resolution of' perhaps?

line 8: The flow of ideas implied by the: 1) <multiple sentences> 2) structure isn't very clear, I'm not sure it's needed

line 10 (and throughout): I found the dating notation used throughout off-putting, personally, but that might just be me. In my experience, the convention I've seen most is that kyr = "thousand years" and ka = "thousand years ago" with 1950 usually taken as the reference for 'ago'. "ka BP" thus feels to me like it's mixing conventions and providing both an 'ago' and a 'before'. I could live with the use of "ka BP" if defined here carefully (and used consistently throughout, but sometimes just "ka" seems to be used to indicate an event date, not a period of time - I'm looking at line 447 as I type this, and table 2 has "kas BP"), but I'm afraid the use of just "a" as the shorthand unit for years (and is that actually meant as 'years before 1950'?), spaced away from the number it's attached to so it can be mistaken for the indefinite article 'a', just grated horribly for me - eg line 455, line 359. It looks OK for eg the SMB numbers as (Gt a^-1) - to avoid confusion perhaps write the full word 'years' in cases like line 359 where "a" is currently used on its own? Related to the date convention, the only place I saw the 1950 reference date stated for the BP 'present' is in the caption to figure 3, and since this isn't a paleoclimate-specific journal I think it would be helpful for the readership to note this much earlier and more plainly.

line 11: having said that melt dominates the changes at 13ka, it would be helpful to say which component causes the increase in SMB at 9ka - is it an increase in melt or decrease in accumulation?

line 12: would be helpful to note the timescale of this AMOC/SMB variability here, it it related to eg Heinrich events or slower/different variations

line 13: not sure a statement on data availability needs to be part of the abstract does it, that intent is implicit in EGU journals now?

line 25: a lack of local horizontal resolution is far from the only reason ESMs struggle with ice sheet SMB - large-scale background climate and circulation biases play a big role, and increasing resolution (horizontal and vertical) certainly doesn't solve everything for either Greenland or Antarctic surface simulations in global models.

line 28: this paragraph is still supposed to be about ice sheets generally, but all these references are for Greenland SMB. You could throw in some Antarctic studies too to be a bit more general - eg Agosta et al., TC 2019

line 29: I spent a while wondering which ice sheets were going to be included in this study, and it's not clarified until much later in the paper. This would be a good opportunity to say this - 'extends the analysis of northern hemisphere SMB changes', perhaps? - and also a good place to note why it's of interest to have a detailed SMB product / analysis for this domain.

line 37: references for what different proxy studies suggest about AMOC variation during these periods would be useful

line 45: I think this statement is too strong - Bauer and Ganopolski show that using fixed parameter values throughout in their simple PDD model does a poor job, but inverting that to say that *only* EBM models can produce realistic ice volume changes seems too much. I'm a little wary as well of the way the term "EBM" might be taken by readers too, whilst we're on the subject - in my (climate modeller) experience it's most often used in the sense of a much-simplified model of radiation balance for an atmospheric column, but here (I think) its being used in a much more general way for any model that explicitly calculates the components of the energy budget and how

they affect temperature and the phase of water at the surface, however complex or simplified. This potential confusion could be eliminated by a disambiguation sentence?

line 53: typos: "a EBM" and "Two kind of simulations"

line 62: this is OK for present-day Greenland (although the accumulation isn't a simple function of height), but should this relationship be taken to be equally strong for all ice sheets, all the time? For example, variation in present-day Antarctica SMB components is more significantly controlled by dynamic atmospheric conditions, rather than a local scaling with topographic height I'd say. Might be worth a caveat here, and a sentence in the discussion about the limitations of this sort of downscaling

line 78: can the intervals/level heights be detailed in a table, maybe supplementary material?

line 84: it would be helpful to note in this list that the phase of precipitation may be changed too

line 105: does the rain that potentially refreezes have a temperature, - perhaps that of the surface - changing the amount of energy required to refreeze it?

line 141: I found it confusing to say "the aging process starts", when so far the only aging process described is for fresh snow. 'Another aging process starts', perhaps? Your aging of the surface albedos is purely a function of time, when quite often the rate of increase of snow-grain size and thus lowering albedo is dependent on temperature and density. Do you know how sensitive your results are to the timescales you've chosen, and whether you would expect them or your results to be different for LGM temperature and accumulation rates?

line 155: Going on with the thread of the previous comment, any table of tuned values almost begs the question of how sensitive your model is, and conclusions are, to the choices you have made, especially when they have been tuned to match one type of climate (the present GrIS) and the model is then applied to very different ones. Do

you have any insight from your undocumented(?) tuning process that you could use to comment on this sensitivity? More work, but maybe possible if you have already done (or could easily do?) the simulations, would be to not only compare your results with MAR SMB for the present, but also for what other models find for GrIS SMB at 2100, or under some other idealised climate forcing. Freely downloadable results do exist for MAR run to 2100 with the climates from various CMIP6 models, from the ISMIP6 preparations. That wouldn't be an evaluation in the same way, but would give your readers some useful reference points for how sensitive your EBM tuning is to the climate it's in.

line 161: I had trouble following this section, but I think I more or less got there in the end. Might be worth thinking about rephrasing. The two cases handled are labelled 1) and 2), but that numbering isn't used again, and it seems like the first case described ("The inflow of snow and ice from above") is actually case 2). How does the temperature profile shift - in proportion to the amount of mass (solid and liquid?) that has been moved between layers? In the case of surface melting (line 165), the mass has not actually left the snowpack until the percolation/refreezing procedure has been completed, so is the change in density profile delayed until that has been done? Although, surface melting lifts the profile /up/, which sounds like melting at the surface makes the top layer /more/ dense than the reference. How does the surface layer ever reduce again to the reference profile, since more snow falling should further increase the layer density (line 163) or be directed to other layers underneath? Is it said anywhere what the top/bottom densities in this profile, or the density of ice in the virtual under-layer, are?

line 190: is even the deep ocean in a true steady state for your LGM initial condition, or are there drifts there to worry about? If it's all a steady state, why are first 5 kyr considered a discardable spin up?

line 195: Freshwater input forced AMOC variability comes up as a feature later - it would be good to mention specifically how glacial runoff/outburst water is treated in

their protocol

line 200: From figure 4 etc, it seems like the ELAs talked about here are not a simple contour on the ice surface demarking where ablation and accumulation balance, but a /potential/ ELA calculatable for the climate in each land surface gridbox regardless of what the actual altitude (or SMB) of the ice sheet was. I'm still not sure what I think that means - precipitation, for instance, is at least partly orographically controlled and either falls in one place or another so it does matter where the surface really is, so not every potential ELA calculated can simultaneously be true...maybe?...but anyway, it would be worth another sentence to explain that the ELA that will be talked about from here on is not a simple contour line on the ice.

line 202: since the ELA is defined as the height where SMB=0 I don't really see what is meant by saying it's less sensitive than SMB

line 232, 242: structurally, do these short paragraphs need their own section headings? Thinking about it, would it make more sense to put the paleo experiment design section later, next to the paleo results, rather than right up front before all the recent history stuff is described and analysed?

line 238: not just cloud, all precipitation in ERA - and thus your EBM-ERAI GrIS accumulation - is also purely a product of their model physics run at global resolution, so I wouldn't expect that to make for a great comparison with the local MAR precipitation, regardless of how well EBM-ERAI can do the surface energy balance.

line 252: from here on, neither refreeze nor runoff is mentioned, only "melt". Since we're often talking about total SMB, I'm not sure whether they really mean surface melt (that might refreeze), or the actual liquid mass leaving the snowpack (runoff). Or are they talking about runoff-precipitation - so the proportion of the original mass that has now been lost? I'm interested to see what their refreezing does anyway, and it's a shame that that's not mentioned, but if they really are talking about just surface melt from here on then I really think the refreezing needs to be shown too.

line 269 (and I elsewhere, like line 316): there's a tendency here to simply blame the biases on in the MPI-ESM runs and the EBM SMB derived from them on inadequate spatial resolution and the local topography. That will undoubtedly be part of it, especially at ice sheet margins, but higher resolution doesn't fix everything in complex climate models, in fact it can make it worse (see Kampenhout et al, TC, 2019 for a very geographically relevant case-study ). Biases in the large-scale climate, inappropriate initialisation, internal variability, missing physics - the list is of course endless, so it's irksomely simplistic to keep saying that the model resolution is too low and no more. It's not like the EBM_MPI-ESM results would = EBM_ERAI if you just ran MPI-ESM at ERAI resolution.

line 275: RACMO hasn't been introduced yet in any way, as a model or an acronym

line 276: You have some tunable control over the rain/snow partitioning in your EBM, so could this MPI-ESM bias be fixed for your purposes?

line 281: I would have thought that larger-scale moisture transports over the ice sheet, which are also quite resolution dependent, would be likely to play a role here too. Low resolution atmospheres often just drizzle too far north and over too-wide areas because they're overly diffusive everywhere, it's not just a local thing you would fix with a higher ice sheet. Again, maybe this is an area where things in the climate models other than simply local grid resolution/topography play a significant role

line 295, 300: I didn't follow the logical thread through here. First MPI-ESM-CR has a higher topography than MAR and the EBM produces too much melt, then later the area in the south is higher than the ISMIP topography and there is less melt. Less melt in which model? Is there a consistent physical link being drawn between these two cases?

line 299: your model has a pretty direct control on how much melting you get when you adjust for a new elevation through the lapse rates - does this point to the lapse rate changes being too high?

[Figure]

line 323: I think this statement on the scope of the paper should be in the Introduction, not all the way back here

line 329: The Antarctic ice sheet has yet to collapse, thankfully

line 332: It would be good to say why these particular time-slices are chosen for analysis.

line 344, 396: the impact of the changes in circulation and the different realisations of AMOC history on the ice sheet SMB sound interesting, it would have been good to hear more about them and how that might have influenced the ice sheet deglaciation. There is some decomposition of whether accumulation or melt is dominating the SMB/ELA trend for the main Glac1D run at different periods - does all this still hold for the ICE6G run? If so, that's an interesting conclusion perhaps that you can say is independent of the ice sheet reconstruction used.

line 394: the final figure of 550 Gt/a doesn't match the values in your table /very/ well. Does the 550 at the end of the transient correspond to the year 1950, rather than ∼2000? Your recent history run does go through 1950 doesn't it, even if only 1980-2010 were analysed - does this 550 match what the historical run did at that point?

line 403-405: references needed for the decline history of these glacial ice sheets

line 450: this information about the meltwater forcing should be in the experiment description

figure 1: is this surface melt, or runoff: the caption says ACC = SMB - MELT? It also says that MAR does not provide accumulation, which I don't understand - MAR gives you snowfall and rainfall? The ELA is just the contour of where the SMB on the surface is 0, isn't it? Is it said in the text why CR has so much more melt in far north

fig 2: (top, middle) to denote the second row and (bottom, middle) to denote the third row is not very clear. Ditto (center, left) and (center, right). With this many panels may just label them a)-p)?

fig 3: Paleoclimate figures often stack different proxies on the same horizontal time axis but offsetting the vertical axes, rather than overlaying everything completely like this - that might work well here too.

fig 4: the 2D ELA surface idea might need explaining in the caption here too. Is the EBM run everywhere? Is there +SMB in the MPI-ESM climates for other places, eg a Siberian ice cap that's just not shown here because masking teh results onto the Glac1D glacier mask?

[Figure]

---

## Referee Comment (RC2) · Anonymous Referee #2 · 30 Sep 2020

Kapsch et al., Analysis of the Surface Mass Balance for Deglacial Climate Simulations

In review in The Cryosphere Discussions

+++++++++++++++++++++++++

This paper is of a potential interest for the geosciences and (paleo-)climate modeling communities.

This study explores the simulated surface mass balance over the Northern Hemisphere ice sheets in the last deglaciation, the last millennium, and the recent past. A special focus is placed on the Greenland mass balance, but the authors also discuss the evolution of surface conditions over the Laurentide and Eurasia ice sheets over the deglaciation.

The manuscript starts with a detailed analysis of how the simulated Greenland climate conditions compare to state-of-the-art regional climate model simulations. Despite the comparatively coarse horizontal resolution, the climate model shows an impressive skill and captures accumulation and ablation patters in the right areas, and with roughly the right magnitude. This comparison gives credibility to the model – at least for simulation of the recent past. The authors then spend some time describing transient paleo-climate simulations from the last glacial maximum to present, and show figures of how the mass balance components of the Laurentide and Eurasian ice sheets evolve in time. The authors also attribute changes in mass balance patterns to changes in the ocean state.

The paper is well written and it presents an interesting and compelling storyline that is of potential interest for the broader (paleo-)climate modeling and cryosphere research communities. I recommend resubmit, subject to some moderate revisions.

+++++++++++++++++++++++++

Major comments:

My only slightly larger comment has to do with a lack of discussion and references of literature on atmospheric circulation changes at the LGM and through the deglaciation. You mention several times that the atmospheric circulation changes over this time period, but no previous studies are cited. Atmospheric circulation changes is arguably not a main focus of your study, but it is a nice gesture to acknowledge work that has been done on this topic in the last several years, not least since it is relevant for your overall modeling approach and for the interpretation of your results.

It is generally accepted that the North Atlantic jet stream and storm track was quite different at the LGM than it is today. Specifically, modeling simulations suggest that the large scale circulation was more zonally oriented than today. Several explanations for this has been proposed, but the most recent explanation (that links all previous interpretations into one theory) of the zonal North Atlantic jet stream and stormtrack is described in: https://www.sciencedirect.com/science/article/abs/pii/S0012821X20300248

A recent overview paper on the PMIP4 LGM simulations: https://cp.copernicus.org/preprints/cp-2019-169/

Circulation changes in the North Atlantic and over Greenland over the last deglaciation: https://agupubs.onlinelibrary.wiley.com/doi/full/10.1002/2017GL074274 https://agupubs.onlinelibrary.wiley.com/doi/full/10.1002/2015GL066042 https://cp.copernicus.org/articles/15/1621/2019/

+++++++++++++++++++++++++

Line comments:

Page 1, line 3: comprehensive -> high complexity?

Page 1, line 5: "downscale atmospheric processes" I assume that you mean radiation?

Page 1, line 6: Here and elsewhere. Maybe more appropriate to say "satellite era" or "recent past" instead of historical period. The latter is used to describe 1850 – present in CMIP6.

Page 1, line 7: "from regional modeling" — add that this is constrained by reanalysis data at the lateral boundaries

Page 1, line 10: Specify that you refer to Northern Hemisphere summer insolation

Page 1, line 19: Here is a relatively recent, comprehensive review that may be worth mentioning in this context: https://agupubs.onlinelibrary.wiley.com/doi/full/10.1029/2018RG000600

Page 2, line 35-36: Good place to mention some work on atmospheric circulation changes in the last deglacial period

Page 2, line 36: "Atlantic Meridional Overturning Circulation"

Page 2, line 42: "Mainly" is repeated. Maybe change the latter to "employed"?

Page 3, line 53: Two kind -> Two kinds

Page 3, line 54: Again, not sure if "historical" is the best word here

Page 3, line 56: SMBs to SMBs — awkward wording

Page 3, line 60: Meaning here is not clear. How do you use an EBM to downscale SMB? Do you mean that the radiation from the EBM is downscaled, or something else?

Page 3, line 60: What influence of clouds? SW radiation? LW radiation?

Page 3, line 70: Meaning of "movement of the snow/ice properties and compaction" is not clear

Page 3, line 71: Explain what elevation classes are

Page 3, line 72: It was technically Lipscomb et al, 2013 that introduced elevation classes in the model https://journals.ametsoc.org/jcli/article/26/19/7352/34179/Implementation-and-Initial-Evaluation-of-the

Page 3, line 80: How is the lapse rate for these factors determined? Also, do you use the same lapse rate in summer and in winter? If yes, is this a valid assumption? What is the sensitivity to this choice?

Page 4, line 85: Not sure if I understand this description. How do you conserve water if this is the case?

Page 4, line 87: This lapse rate is quite a bit lower than the ICAO value of -6.5 K/km. Explain this choice, and did you test the sensitivity to this value?

Page 4, line 97: Technical detail but what happens to (latent and sensible) energy fluxes if the atmosphere model simulates liquid precipitation, but the height corrected

temperature is below freezing (and vice versa)?

Page 5, line 136: It is a bit clunky to define WE at the end of the sentence. Can this definition be moved earlier in the sentence?

Page 5, line 141: brighten -> increase (?)

Page 6, line 165: "Once..." — Meaning here is not clear. Do you mean that it is removed?

Page 6, line 180: Technical detail, but please specify that this is the horizontal resolution

Page 7, line 189: Acronym "ka BP" is not defined

Page 7, line 195: Not sure if I fully understand this modeling strategy. Do you run 10 years with a constant forcing before updating the boundary conditions, or do you advance the orbital clock, topography, etc. by 10 years every model year?

Page 7, line 196: What happens to vegetation in areas that are deglaciated?

Page 7, line 203: "it is a good proxy" –> "hence, it is a proxy for..."

Page 8, line 214: " for a long enough adjustment" is a bit colloquial

Page 8, line 232: Why did you use ERA-Interim instead of the newer ERA5? Do you expect different results with a newer reanalysis product?

Page 9, line 250: ...historical climate conditions -> recent past

Page 9, line 254: remind the reader that this is the "coarse" and "low resolution" simulations

Page 9, line 264: Say something about how these numbers change over Greenland (latitudes of interest). Should be a factor of 2-3 difference from the equator

Page 10, line 286: Good reference https://tc.copernicus.org/articles/13/1547/2019/

Page 10, line 287: Sentence starting "The overestimation" is a bit clunky and can be simplified

Page 11, line 316: What figure(s) are you discussion here?

Page 12, line 344: Good place to cite some work on atmospheric dynamics /circulation in the deglaciation. See major comment

Page 12, line 354: shrinks -> recedes

Page 12, line 357: Further, it points towards the fact -> Further, it suggests

Page 13, line 380: typo: his -> its (?)

Page 14, line 416: Even though the AMOC probably plays an important role for this response, it is the atmosphere that primarily interacts with the ice sheets. I would suggest extending this discussion with changes in the atmospheric circulation in mind, and perhaps cite a few papers that have looked at these interactions before

Page 15, line 467: What about regional and large scale atmospheric circulation?

Page 15, line 468: You could cite this paper when talking about other feedback processes: https://agupubs.onlinelibrary.wiley.com/doi/full/10.1029/2018RG000600

Page 16, line 491: Can you end with a slightly more comprehensive future outlook? Where do you want to take this in the future, and how will this new modeling capability be used in e.g. simulations of the future climate evolution, and/or other paleo-climate states?

Table 2: This simulation contributed to CMIP6

Figure 1: Here and elsewhere. Spectral colorbars are bad for people with color blindness. Please us a non-spectral color scale if possible

Figure 4: Here and elsewhere. The SMB colorscale is a but crowded. If possible, use fewer intervals.

[Figure]

---

## Author Comment (AC1) · 19 Nov 2020

**Response to the review of the manuscript "Analysis of the Surface Mass Balance for Deglacial Climate Simulations" submitted to The Cryosphere.**

We thank both reviewers for their comprehensive reviews and very useful suggestions to improve the manuscript. We have addressed all of their comments and believe that the changes will significantly improve the manuscript. In the following, reviewer comments are highlighted in blue, author responses in black.

Anonymous Referee #1

General comments
I did enjoy reading this, and it feels like I've spent a lot of time writing it up – apologies for the lateness - I found lots of small things to ask about, see below.
We thank the reviewer for his very comprehensive review and are glad for the very good suggestions to improve the motivation and framing of the manuscript.

I felt there was one wider, general issue that hopefully would be easy enough to address - as I got to the paleoclimate section I didn't think that a solid motivation had really been given for /why/ a simulation of deglacial SMB was being done, and thus I wasn't really sure what to be looking for in how it's been described and analysed. Even by the end this isn't clear, as the Conclusions don't conclude anything that sounds obviously new about the glacial climate system or the model itself. In a couple of places it's said that the SMB fields will be made available to the ice sheet community for their use, but the first part of section 4 says that the simulation isn't going be evaluated, so I don't really buy this as sufficient motivation for the whole exercise - are these modelling groups going to want to use a new SMB product from a climate simulation that hasn't been evaluated for how it compares to evidence of what actually happened?
We agree that the motivation of this analysis did not come out clearly in the manuscript. We changed the abstract and introduction to state our motivation more clearly. The main motivation for our study is that for long term studies of ice sheet changes (fully coupled climate-ice sheet or ice sheets model only simulations) realistic SMBs derived from long Earth System Model (ESM) simulations are needed. Here, we present an EBM that is used to downscale coarse resolution ESM output. We evaluate the EBM and apply it to the first transient simulations of the last deglaciation with the MPI-ESM earth system model and prescribed ice sheets. A misunderstanding is that these simulations are not evaluated. We are currently undertaking an analysis regarding the sensitivity of the experiments to differences in the boundary forcing (Glac1D vs Ice6G). A publication of the results is expected in the near future.

There is some interesting decomposition of how accumulation and melt factors balance differently in different periods and how the (perhaps non-obvious) relationship between total SMB and ELA changes too, but only for GrIS - is it different in different regions? - and no attempt is made at general statements about the global implications or testable hypotheses for underlying principles. Personally, I wanted to know more about how some of the details of the snow physics were being forced by the climate through the deglaciation - albedo, refreezing etc, rather than just the resulting SMB, but that's just one idea.
We shortly discuss these changes in the section for the Laurentide and Fennoscandian ice sheets. However, there are different processes acting across these ice sheets as they respond to both changes in the Atlantic and Pacific. So the relationships are not uniform over the ice sheets. We believe that part of this comment is also related to the confusion of the melt definition (see comment to line 252) - we now defined the melt explicitly in the manuscript, which includes refreezing processes. We also revised the introduction and specifically the motivation for this study, which is to simulate a realistic SMB for long-term studies of ice sheet and climate interactions. Thereby, we hope to have addressed the concerns of the reviewer.

I think the authors should either decide on a clear main goal that's stated at the front of the paper and runs all the way through, or be more clear about the separate aims of each part - either way that should make it easier to see what the paper as a whole is working towards, and clearer what should be in the final conclusions.

We revised the introduction and parts of the conclusion to emphasize our motivation and the aim of the present paper.

Detailed comments:
line 4: "deglacial climate" - quite specifically the /last/ deglaciation

We changed it to 'the last deglaciation'

line 5: "allows to resolve" isn't grammatically great - 'allows the resolution of' perhaps?

Thanks, we changed this.

line 8: The flow of ideas implied by the: 1) <multiple sentences> 2) structure isn't very clear, I'm not sure it's needed

We removed this and rephrase slightly.

line 10 (and throughout): I found the dating notation used throughout off-putting, personally, but that might just be me. In my experience, the convention I've seen most
is that kyr = "thousand years" and ka = "thousand years ago" with 1950 usually taken as the reference for 'ago'. "ka BP" thus feels to me like it's mixing conventions and providing both an 'ago' and a 'before'. I could live with the use of "ka BP" if defined here carefully (and used consistently throughout, but sometimes just "ka" seems to be used to indicate an event date, not a period of time - I'm looking at line 447 as I type this, and table 2 has "kas BP"), but I'm afraid the use of just "a" as the shorthand unit for years (and is that actually meant as 'years before 1950'?), spaced away from the number it's attached to so it can be mistaken for the indefinite article 'a', just grated horribly for me - eg line 455, line 359. It looks OK for eg the SMB numbers as (Gt a^-1) - to avoid confusion perhaps write the full word 'years' in cases like line 359 where "a" is currently used on its own? Related to the date convention, the only place I saw the 1950 reference date stated for the BP 'present' is in the caption to figure 3, and since this isn't a paleoclimate-specific journal I think it would be helpful for the readership to note this much earlier and more plainly.

Thank you very much for the clarification. We revised the manuscript in respect to the notation mentioned here and introduce the reference year (1950) earlier in the manuscript.

line 11: having said that melt dominates the changes at 13ka, it would be helpful to say which component causes the increase in SMB at 9ka - is it an increase in melt or decrease in accumulation?

We added 'After 13 ka the increase in melt begins to dominate the SMB decrease'.

line 12: would be helpful to note the timescale of this AMOC/SMB variability here, it it related to eg Heinrich events or slower/different variations

We will add 'centennial-scale episodes'.

line 13: not sure a statement on data availability needs to be part of the abstract does it, that intent is implicit in EGU journals now?

It is not and we removed it from the abstract. Thanks for the suggestion.

line 25: a lack of local horizontal resolution is far from the only reason ESMs struggle with ice sheet SMB - large-scale background climate and circulation biases play a big role, and increasing resolution (horizontal and vertical) certainly doesn't solve everything for either Greenland or Antarctic surface simulations in global models.

We do agree with this point and mention these shortcomings by including *"This is specifically challenging, as ESMs exhibit biases and the horizontal resolution is often not sufficient to capture small scale climate features, e.g., sharp topographic gradients at the ice sheet margins as well as cloud, snow and firn processes (e.g. Lenaerts et al. 17, van Kampenhout et al 2017, Fyke et al. 2018)."*

line 28: this paragraph is still supposed to be about ice sheets generally, but all these references are for Greenland SMB. You could throw in some Antarctic studies too to be a bit more general - eg Agosta et al., TC 2019
We added some references about SMB changes over Antarctica (Lenaerts et al., 2012b, van Wessem et al., 2018).

line 29: I spent a while wondering which ice sheets were going to be included in this study, and it's not clarified until much later in the paper. This would be a good opportunity to say this - 'extends the analysis of northern hemisphere SMB changes', perhaps?
- and also a good place to note why it's of interest to have a detailed SMB product / analysis for this domain.
Thank you for your suggestion! We included that we are analyzing the northern hemisphere. And we add a short paragraph why we need realistic SMBs specifically for this region and time period. We added in the end of the paragraph: *"These climate changes and the variability associated with the changes in the northern hemispheric ice sheets during the deglaciation and the resemblance to the expected future climate change emphasize the need for a realistic representation of the SMB for past and future stand-alone ice-sheet and coupled climate-ice-sheet model simulations (Fyke et al., 2018)."*

line 37: references for what different proxy studies suggest about AMOC variation during these periods would be useful
We added some references (Heinrich 1988, Keigwin et al 1994, Vidal et al 1997).

line 45: I think this statement is too strong - Bauer and Ganopolski show that using fixed parameter values throughout in their simple PDD model does a poor job, but inverting that to say that *only* EBM models can produce realistic ice volume changes seems too much. I'm a little wary as well of the way the term "EBM" might be taken by readers too, whilst we're on the subject - in my (climate modeller) experience it's most often used in the sense of a much-simplified model of radiation balance for an atmospheric column, but here (I think) its being used in a much more general way for any model that explicitly calculates the components of the energy budget and how they affect temperature and the phase of water at the surface, however complex or simplified. This potential confusion could be eliminated by a disambiguation sentence?
We use the term EBM more specifically as synonym for a model that is used to calculate and downscale the surface mass balance from ESMs independent of its complexity. We are excluding PDD models, which use statistical relationships between melt and temperature patters based on present-day observations. We changed the sentence in response to this review and reformulated the motivation for this study.

line 53: typos: "a EBM" and "Two kind of simulations"
Thanks for pointing these out!

line 62: this is OK for present-day Greenland (although the accumulation isn't a simple function of height), but should this relationship be taken to be equally strong for all ice sheets, all the time? For example, variation in present-day Antarctica SMB components is more significantly controlled by dynamic atmospheric conditions, rather than a local scaling with topographic height I'd say. Might be worth a caveat here, and a sentence in the discussion about the limitations of this sort of downscaling

We added a note in the paragraph that emphasizes that the elevation dependence is not all there is: *"Note, that an elevation dependence of the SMB components is a simplified assumption and valid mainly for present-day Greenland. Atmospheric dynamics also significantly contribute to variations in the SMB components, specifically over present-day Antarctica."*

line 78: can the intervals/level heights be detailed in a table, maybe supplementary material?
We added a Table to the supplementary material.

line 84: it would be helpful to note in this list that the phase of precipitation may be changed too
We added: *"Total precipitation rates (liquid and solid) are corrected under consideration of the height-desertification effect. This halves the precipitation for an orography height difference of 1000 m above a threshold height of 2000 m for each grid point (Budd and Smith, 1979). Note, that snowfall is determined from the total precipitation for height corrected near-surface air temperatures below 0C within the EBM."*

line 105: does the rain that potentially refreezes have a temperature, - perhaps that of the surface - changing the amount of energy required to refreeze it?
We added *"The temperature of rain is assumed to be equal to the height corrected near-surface air temperature."* for clarity.

line 141: I found it confusing to say "the aging process starts", when so far the only aging process described is for fresh snow. 'Another aging process starts', perhaps? Your aging of the surface albedos is purely a function of time, when quite often the rate of increase of snow-grain size and thus lowering albedo is dependent on temperature and density. Do you know how sensitive your results are to the timescales you've chosen, and whether you would expect them or your results to be different for LGM temperature and accumulation rates?
Here, it is not another aging process but the aging of the albedos as described above. Maybe 'the aging process starts from the beginning' would clarify? The parametrization used here goes back to Oerlemans and Knap (1998). We did not do a comprehensive testing regarding the sensitivity to the length scales, as we mainly followed the parametrization in the above mentioned reference. Also computational limitations for the long-term simulations make several attempts with different tuning factors not feasible.

line 155: Going on with the thread of the previous comment, any table of tuned values almost begs the question of how sensitive your model is, and conclusions are, to the choices you have made, especially when they have been tuned to match one type of climate (the present GrIS) and the model is then applied to very different ones. Do you have any insight from your undocumented(?) tuning process that you could use to comment on this sensitivity? More work, but maybe possible if you have already done (or could easily do?) the simulations, would be to not only compare your results with MAR SMB for the present, but also for what other models find for GrIS SMB at 2100, or under some other idealised climate forcing. Freely downloadable results do exist for MAR run to 2100 with the climates from various CMIP6 models, from the ISMIP6 preparations. That wouldn't be an evaluation in the same way, but would give your readers some useful reference points for how sensitive your EBM tuning is to the climate it's in.
The SMB derived here is sensitive to all of these tuning factors, which is the reason why we chose to present the values in the Table. As for any tunable variable in a model we made the choice between being as close as possible to present-day observations and model performance. Due to computational limitations for the long-term simulations we have chosen one set of parameter for the deglacial simulations, hence, we do not know how sensitive the

choice is under a changing climate. That being said, we are not confident that a comparison of the SMBs in a future climate with MAR will allow us to fully evaluate the obtained SMBs for a changing climate. It will be difficult to disentangle the effects on the SMB that arise from differences in the climate forcing itself (MPI-ESM vs. MAR forcing). However, we applied the EBM to a simulation with the MPI-ESM high resolution setup for the SMBMIP inter-comparison and the results showed that the setup was specifically good in representing the trends of the Greenland mass loss between 2003-2012, indicating that not just the SMB mean climate over Greenland, but also changes in the climate are represented reasonably well as compared to observations and RCMs (see Fettweis et al., 2020). We added the following information to the manuscript: *"The same parameters were applied in an EBM simulation forced with output from a high resolution MPI-ESM simulation for historical climate conditions within the scope of a SMB model inter-comparison (SMBMIP; Fettweis et al. 2020). The results showed that the derived SMBs were very similar to observations in terms of the SMB mean climate as well as the SMB trend (2003-2012)."*

line 161: I had trouble following this section, but I think I more or less got there in the end. Might be worth thinking about rephrasing. The two cases handled are labelled 1) and 2), but that numbering isn't used again, and it seems like the first case described ("The inflow of snow and ice from above") is actually case 2). How does the temperature profile shift - in proportion to the amount of mass (solid and liquid?) that has been moved between layers? In the case of surface melting (line 165), the mass has not actually left the snowpack until the percolation/refreezing procedure has been completed, so is the change in density profile delayed until that has been done? Although,surface melting lifts the profile /up/, which sounds like melting at the surface makes the top layer /more/ dense than the reference. How does the surface layer ever reduce again to the reference profile, since more snow falling should further increase the layer density (line 163) or be directed to other layers underneath? Is it said anywhere what the top/bottom densities in this profile, or the density of ice in the virtual under-layer, are?

We have revised the complete paragraph for clarification under consideration of the questions mentioned here. The procedure applied here is instantaneous. We first calculate compaction and then surface melt. Percolation and refreezing can affect the density profile. The temperature profile follows the movement of mass. We also tried to clarify how the density profile changes for different conditions. The top and bottom densities are those of ice and snow. We hope that the rewritten section allows for a better understanding of the used parametrization.

line 190: is even the deep ocean in a true steady state for your LGM initial condition, or are there drifts there to worry about? If it's all a steady state, why are first 5 kyr considered a discardable spin up?

Note that these are transient simulations of the last deglaction, hence, there is no steady-state. We initialized the simulation from a spun-up glacial steady-state at 26 ka. We are only investigating the last 21 kyrs for the model to adjust to the changes and because this is the most interesting time period. 5000 years are enough to capture the drift in the deep ocean. Therefore, we considered the rest as additional spin-up. As this seems to confusing we rephrased.

line 195: Freshwater input forced AMOC variability comes up as a feature later - it would be good to mention specifically how glacial runoff/outburst water is treated in their protocol

We added following information: *"Freshwater from melting ice sheets is calculated from the volume changes in the ice sheet reconstructions for each grid point. For grid cells over land melt water is distributed through the hydrological discharge model, over ocean it is discharged into the adjacent ocean grid cells. "*

line 200: From figure 4 etc, it seems like the ELAs talked about here are not a simple contour on the ice surface demarking where ablation and accumulation balance, but a /potential/ ELA calculatable for the climate in each land surface gridbox regardless of what the actual altitude (or SMB) of the ice sheet was. I'm still not sure what I think that means - precipitation, for instance, is at least partly orographically controlled and either falls in one place or another so it does matter where the surface really is, so not every potential ELA calculated can simultaneously be true...maybe?...but anyway, it would be worth another sentence to explain that the ELA that will be talked about from here on is not a simple contour line on the ice.
We included a sentence that clarifies the ELA used in this study, which is indeed a potential ELA, as it is calculated in each grid point. *"At heights above the ELA  it is thermodynamically possible to accumulate snow throughout the year and form an ice sheet or glacier. At elevations below the ELA melt dominates accumulation and no ice sheet can form. Here, the ELA is calculated in each grid point, hence, resembles a potential ELA. It is a proxy for climate changes affecting the ice sheets. "*

line 202: since the ELA is defined as the height where SMB=0 I don't really see what is meant by saying it's less sensitive than SMB
To obtain realistic SMBs we need to downscale the SMB onto a different grid (to the ice sheet topography from the reconstructions), while the ELA can be determined directly on the atmospheric model grid. The downscaled SMB shows enhanced melt patterns specifically at the margins of the ice sheet where the simulated climate from the coarse resolution model and the high resolution ice sheet topography potentially do not fit together (in a fully coupled simulation an ice sheet would likely not survive in these areas). If these values are integrated, the resulting SMB may be underestimated. Integrating the ELA on the native model grid allows for a more model consistent representation of patterns. Hence, it is less sensitive than the SMB. We rephrased this sentence to make it more clear. *"As the ELA estimate is calculated on the native model grid it is more consistent with the model physics and boundary conditions used in the simulations than the downscaled SMB. Hence, integrated values of the ELA are less sensitive to changes in the ice sheet mask than those of the SMB. "*

line 232, 242: structurally, do these short paragraphs need their own section headings? Thinking about it, would it make more sense to put the paleo experiment design section later, next to the paleo results, rather than right up front before all the recent history stuff is described and analysed?
These are good points but we have decided to put it this way around, as our historical simulations are branched off from the deglaciation experiment. So to describe the recent history simulations we believe we need to describe how we initialized them. To avoid too much repetition we therefore decided to put it this way around. We changed the sub headings to "Evaluation Data" to avoid too many subtitles and hope that this is sufficient.

line 238: not just cloud, all precipitation in ERA - and thus your EBM-ERAI GrIS accumulation - is also purely a product of their model physics run at global resolution, so I wouldn't expect that to make for a great comparison with the local MAR precipitation, regardless of how well EBM-ERAI can do the surface energy balance.
We fully agree with the reviewer. This sentence was meant for readers not fully aware of the limitations of reanalysis. We added some examples (precipitation, clouds).

line 252: from here on, neither refreeze nor runoff is mentioned, only "melt". Since we're often talking about total SMB, I'm not sure whether they really mean surface melt (that might refreeze), or the actual liquid mass leaving the snowpack (runoff). Or are they talking about runoff-precipitation - so the proportion of the original mass that has now been lost? I'm interested to see what their refreezing does anyway, and it's a shame that that's not

mentioned, but if they really are talking about just surface melt from here on then I really think the refreezing needs to be shown too.
We believe that we have introduced melt in the introduction *" The SMB is determined by mass gain due to accumulation, as a result of snow deposition, and mass loss by ablation, induced by thermodynamical processes at the surface and subsequent melt-water runoff (Ettema et al., 2009)".* Here the melt is defined as the mass leaving the snowpack, that means melt water that refreezes is not considered in this estimate. We added a sentence to clarify this definition. *"In the following, accumulation is defined as mass gain due to snow deposition and melt as mass loss due to ablation (often referred to as runoff). Refreezing processes are considered in the melt estimate."*

line 269 (and I elsewhere, like line 316): there's a tendency here to simply blame the biases on in the MPI-ESM runs and the EBM SMB derived from them on inadequate spatial resolution and the local topography. That will undoubtedly be part of it, especially at ice sheet margins, but higher resolution doesn't fix everything in complex climate models, in fact it can make it worse (see Kampenhout et al, TC, 2019 for a very geographically relevant case-study ). Biases in the large-scale climate, inappropriate initialisation, internal variability, missing physics - the list is of course endless, so it's irksomely simplistic to keep saying that the model resolution is too low and no more. It's not like the EBM_MPI-ESM results would = EBM_ERAI if you just ran MPI-ESM at ERAI resolution.
We are aware that model resolution is not the only reason for the discrepancies between MAR, EBM_ERAI and MPI-ESM-LR and MPI-ESM-CR. In fact, there are many uncertainties in our system, related to biases in the model climate (clouds, precipitation, surface processes, ...), the parametrization used to downscale the SMB (including e.g. a constant lapse rate over the whole ice sheet) and of course the vertical and horizontal interpolations from the elevation classes on coarse resolution to the high resolution ice sheet topography. We revised the manuscript throughout and added specifically *"The comparison between EBM_MPI-ESM-LR and EBM_MPI-ESM-CR indicates that an increase in resolution cannot resolve all biases. This is in line with findings by Kampenhout et al. (2019), showing that a regional grid refinement in simulations with the Community Earth System Model (CESM) did not improve all SMB components. Model biases, e.g. in the large-scale circulation, clouds and precipitation patterns (Mauritsen et al., 2019), as well as uncertainties due to internal variability are exhibited in all ESMs and explain part of the differences seen in the presented comparison."* and also revised other places throughout the manuscript (e.g. line 316). We also believe that we have discussed other possibilities in the discussion section: *"A comparison with SMBs derived from a simulation with the MPI-ESM-LR setup and ERA-Interim reanalysis reveal that discrepancies between the SMBs derived from MPI-ESM-CR and MAR are a result of the coarse resolution of the model (e.g., the extensive melt in the North of the Greenland ice sheet) and due to the quality of the forcing itself (e.g., precipitation patterns). In contrast to ERA-Interim, MPI-ESM evolves freely and does not assimilate any surface observations; hence, differences are to be expected. Further differences are related to underlying topographies of the native models as well as the fact that most fluxes within the EBM are parameterized, as they are not directly available from the model simulations in the required temporal resolution."*

line 275: RACMO hasn't been introduced yet in any way, as a model or an acronym
Thanks, we introduced the acronym here.

line 276: You have some tunable control over the rain/snow partitioning in your EBM, so could this MPI-ESM bias be fixed for your purposes?
Precipitation is considered as snow if the height corrected near-surface temperatures are below 0C. This would be of course a tunable value, but we do not believe that it is very physical to change this relationship.

line 281: I would have thought that larger-scale moisture transports over the ice sheet,

which are also quite resolution dependent, would be likely to play a role here too. Low
resolution atmospheres often just drizzle too far north and over too-wide areas because
they're overly diffusive everywhere, it's not just a local thing you would fix with a higher
ice sheet. Again, maybe this is an area where things in the climate models other than
simply local grid resolution/topography play a significant role

This is likely also a contribution but we believe it is rather remarkable how differences in the
topography match the differences in the precipitation. This points to the topography being
the dominant driver of those difference, but does not rule out other processes. As we added
a couple of sentences in response to comment line 269 we believe we have covered this
point.

line 295, 300: I didn't follow the logical thread through here. First MPI-ESM-CR has a higher
topography than MAR and the EBM produces too much melt, then later the area in the south
is higher than the ISMIP topography and there is less melt. Less melt in which model? Is
there a consistent physical link being drawn between these two cases?

We apologize for the confusion. The first sentence should read *"has a higher topography
than ISMIP..."* - as is shown in Fig. 2. We believe that we explained the physical
mechanisms by *"One problem of downscaling melt in these regions is that temperatures are
always at the melting point during melting. By projecting the temperatures onto lower
elevations, the height corrected temperatures depart significantly from the melting point
towards higher temperatures. Hence, the vertical downscaling from higher elevations to low
elevations overestimates melting"* and that the reason for the confusion lies in the small
error.

line 299: your model has a pretty direct control on how much melting you get when you
adjust for a new elevation through the lapse rates - does this point to the lapse rate changes
being too high?

This is a good question. The lapse rate is used in the model as tuning parameter and we
have chosen a value that results in realistic SMBs for the whole Greenland ice sheet. In the
current version of the model it is considered as relatively low (4.6K/km), which certainly is at
the lower end for Greenland temperatures. It points towards the challenges in using one
constant lapse rate over all ice sheets and certainly is a shortcoming in our EBM.

line 323: I think this statement on the scope of the paper should be in the Introduction,
not all the way back here

We moved this sentence up in the introduction. Thanks for the suggestion.

line 329: The Antarctic ice sheet has yet to collapse, thankfully

We fully agree and added 'all northern hemispheric ice sheets'.

line 332: It would be good to say why these particular time-slices are chosen for analysis.

These time slices show the most significant changes during the deglaciation. We added this
information. "... in order to indicate the most drastic changes in the northern hemispheric ice
sheet configuration."

line 344, 396: the impact of the changes in circulation and the different realisations of
AMOC history on the ice sheet SMB sound interesting, it would have been good to hear
more about them and how that might have influenced the ice sheet deglaciation. There
is some decomposition of whether accumulation or melt is dominating the SMB/ELA
trend for the main Glac1D run at different periods - does all this still hold for the ICE6G
run? If so, that's an interesting conclusion perhaps that you can say is independent of
the ice sheet reconstruction used.

Good point! We added some information on the atmospheric circulation changes (see also
comments by reviewer #2). We also see similar melt-accumulation relationship for the Ice6G
simulation, although the absolute magnitude differs. Melt is close to zero until about 14 ka

and also reduces during the major AMOC slowdowns. Note, that the AMOC slowdowns are somewhat different in their timings and occurrence, due to uncertainties in the reconstructions - something we are currently investigating in a separate study. As the relationships hold we included a sentence pointing this out. Thanks for this suggestion.

line 394: the final figure of 550 Gt/a doesn't match the values in your table /very/ well. Does the 550 at the end of the transient correspond to the year 1950, rather than _2000? Your recent history run does go through 1950 doesn't it, even if only 1980-2010 were analysed - does this 550 match what the historical run did at that point?
The historical simulation is somewhat different from the transient simulation, as it includes more realistic forcings (e.g. volcanoes, land use, anthropogenic forcings). Therefore the SMB values at the end of the transient simulation (1950) are higher than in the beginning of the historical simulations. The differences in the forcing are also the reason why we performed the last millenniums simulation (starting at 1850 from the deglaciation experiment), in order for the model to adapt to the changes in the forcings. Hence, we have no overlap between the end of the deglaciation and the historical simulation to directly compare the values. We revised the sentence to *"These values are similar to values observed during the 21st-century, although slightly higher as no anthropogenic forcings are considered in the deglaciation simulation (see Fig. 4, Section 3 and Table 3)"* for clarification.

line 403-405: references needed for the decline history of these glacial ice sheets
We added references to this section.

line 450: this information about the meltwater forcing should be in the experiment description
As we included how we determine the freshwater flux from the ice sheet reconstructions in the experiment section in response to an earlier comment we believe that this is covered. The melt water pulse is prescribed by the reconstructions, which are introduced in the experimental description.

figure 1: is this surface melt, or runoff: the caption says ACC = SMB - MELT? It also says that MAR does not provide accumulation, which I don't understand - MAR gives you snowfall and rainfall? The ELA is just the contour of where the SMB on the surface is 0, isn't it? Is it said in the text why CR has so much more melt in far north
Accumulation in our study is defined as the mass gain due to snowfall. While MAR provides snowfall and rainfall it does not explicitly provide how much of the precipitation accumulates over the ice sheet. We believe that calculating it as residual is therefore the right thing to do. We believe that melt in the north of the ice sheet is a result of the underlying topography (and to some extent probably also model biases) and cite from the text: *"Comparisons with EBM_MPI-ESM-LR, which shows less melt in the north and west of the ice sheet as compared to EBM_MPI-ESM-CR (Fig. 1 and 2), confirm that differences in the melt patterns are linked to the underlying topographies of the model versions. MPI-ESM-LR is slightly higher than MPI-ESM-CR and thereby closer to MAR on the northern and western flanks of the ice sheet, hence EBM_MPI-ESM-LR$ shows less melt than EBM_MPI-ESM-CR in these areas."*

fig 2: (top, middle) to denote the second row and (bottom, middle) to denote the third row is not very clear. Ditto (center, left) and (center, right). With this many panels may just label them a)-p)?
We added labels to Fig. 1 and Fig. 2. Thanks for the suggestion.

fig 3: Paleoclimate figures often stack different proxies on the same horizontal time axis but offsetting the vertical axes, rather than overlaying everything completely like this - that might work well here too.

Thanks for the suggestions. We are aware of these plots but think that in the current presentation relationships between e.g. the ELA and CO2 as well as the MOC are easier to see and interpret. Similarly for melt and accumulation relationships.

fig 4: the 2D ELA surface idea might need explaining in the caption here too. Is the EBM run everywhere? Is there +SMB in the MPI-ESM climates for other places, eg a Siberian ice cap that's just not shown here because masking teh results onto the Glac1D glacier mask?

We clarified the ELA definition and that we masked the SMB and ELA with the glacier mask by including "*The SMB is interpolated on the Glac1D topography for each individual time slice and masked with the Glac1D glacier mask. The ELA, defined as elevation where the SMB equals zero, is calculated for each grid point on the native MPI-ESM-CR model grid from the 3-D SMB in each grid point (see Sections 2.1 and 2.3). The ELA is masked with the glacier mask used in the MPI-ESM-CR simulations.*"

---

## Author Comment (AC2) · 19 Nov 2020

**Response to the review of the manuscript "Analysis of the Surface Mass Balance for Deglacial Climate Simulations" submitted to The Cryosphere.**

We thank both reviewers for their comprehensive reviews and very useful suggestions to improve the manuscript. We have addressed all of their comments and believe that the changes will significantly improve the manuscript. In the following, reviewer comments are highlighted in blue, author responses in black.

Anonymous Referee #2

Major comments:
My only slightly larger comment has to do with a lack of discussion and references of literature on atmospheric circulation changes at the LGM and through the deglaciation. You mention several times that the atmospheric circulation changes over this time period, but no previous studies are cited. Atmospheric circulation changes is arguably not a main focus of your study, but it is a nice gesture to acknowledge work that has been done on this topic in the last several years, not least since it is relevant for your overall modeling approach and for the interpretation of your results. It is generally accepted that the North Atlantic jet stream and storm track was quite different at the LGM than it is today. Specifically, modeling simulations suggest that the large scale circulation was more zonally oriented than today. Several explanations for this has been proposed, but the most recent explanation (that links all previous interpretations into one theory of the zonal North Atlantic jet stream and stormtrack is described in:
https://www.sciencedirect.com/science/article/abs/pii/S0012821X20300248
A recent overview paper on the PMIP4 LGM simulations:
https://cp.copernicus.org/preprints/cp-2019-169/
Circulation changes in the North Atlantic and over Greenland over the last deglaciation: https://agupubs.onlinelibrary.wiley.com/doi/full/10.1002/2017GL074274
https://agupubs.onlinelibrary.wiley.com/doi/full/10.1002/2015GL066042
https://cp.copernicus.org/articles/15/1621/2019/
We fully agree with the reviewer and thank him for referring to the literature. The atmospheric circulations plays an important role in our simulations and we find similar changes in the atmospheric circulation during the LGM, as described in Löfverström and Lora (2017). In a study currently under preparation for publication, we also find that uncertainties in the ice sheet reconstructions lead to significant differences in the atmospheric circulation. As we focus on the SMB and drivers of SMB changes in the current manuscript, we will mainly touch on these processes in the revised manuscript (see detailed response below).

++++++++++++++++++++++++++
Line comments:
Page 1, line 3: comprehensive -> high complexity?
We change to state-of-the-art as we believe that these models are both comprehensive (in their sub-model components) and have a high complexity.

Page 1, line 5: "downscale atmospheric processes" I assume that you mean radiation?
In this case we downscale the energy balance at the surface, more specifically the SMB, accumulation and melt, not just radiation. We changed the sentences to *"An energy balance model (EBM) is used to calculate and downscale the SMB on higher spatial resolution and allows the resolution of SMB variations due to topographic gradients not resolved by the ESM"* for clarity.

Page 1, line 6: Here and elsewhere. Maybe more appropriate to say "satellite era" or "recent past" instead of historical period. The latter is used to describe 1850 – present in CMIP6.

This is correct, but they are also labeled as historical in the CMIP experiments. We believe that defining the years of our analysis and introducing the period as historical is sufficient here.

We did not add this in the abstract, but into the introduction, as it is an important information that we did not mention throughout the manuscript. Thanks for the suggestion.

Page 1, line 10: Specify that you refer to Northern Hemisphere summer insolation
Thanks for pointing this out. We changed this.

Page 1, line 19: Here is a relatively recent, comprehensive
review that may be worth mentioning in this context:
https://agupubs.onlinelibrary.wiley.com/doi/full/10.1029/2018RG000600
Thanks for pointing us to this review. We added the reference to this review in the beginning of this section in response to another comment.

Page 2, line 35-36: Good place to mention some work on atmospheric circulation changes in the last deglacial period
Thanks. As mentioned before we extended this paragraph and included some information on circulation and climate changes as well as some of the proposed references: *"The collapse of the ice sheets also resulted in significant changes in the atmospheric and oceanic circulation as well as associated climate features (e.g. Lofverstrom et al. 2017). Orographic changes, induced by the decrease of the Laurentide and Cordilleran ice sheets, led to changes in the Northern Hemispheric stationary waves and thereby the North Atlantic jet stream, which significantly affected the northern hemispheric climate (e.g. changes in precipitation and temperature patterns; Andres and Tarasov, 2019; Lofverstrom et al. 2020, Kageyama et al. 2020)."*

Page 2, line 36: "Atlantic Meridional Overturning Circulation"
Thanks. We will change this.

Page 2, line 42: "Mainly" is repeated. Maybe change the latter to "employed"?
Thanks! Changed.

Page 3, line 53: Two kind -> Two kinds
Thanks. Changed.

Page 3, line 54: Again, not sure if "historical" is the best word here
Please see comment above.

Page 3, line 56: SMBs to SMBs — awkward wording
We rephrased to *"...compare obtained SMBs to output from the regional climate model MAR".*

Page 3, line 60: Meaning here is not clear. How do you use an EBM to downscale SMB? Do you mean that the radiation from the EBM is downscaled, or something else?
We rephrased the sentence to *"We use an EBM to calculate and downscale"*... Surface fields from the MPI-ESM simulation are used to calculate the SMB, which is then interpolated on a higher resolution ice sheet topography. Note that we not just correct radiation but also precipitation, pressure, etc.

Page 3, line 60: What influence of clouds? SW radiation? LW radiation?

We are not sure what is meant here, as clouds are not mentioned in this sentence. We assume that the reviewer is talking about Line 69, where we can add *"The main improvements are 1) an advanced broadband albedo scheme considering aging, snow depth dependency, and the influence of the cloud coverage on the thermal radiation,..."*

Page 3, line 70: Meaning of "movement of the snow/ice properties and compaction" is not clear

We rephrased to "the consideration of snow compaction and the vertical advection of snow/ice properties".

Page 3, line 71: Explain what elevation classes are
Page 3, line 72: It was technically Lipscomb et al, 2013 that introduced elevation classes in the model https://journals.ametsoc.org/jcli/article/26/19/7352/34179/Implementation-and-Initial-Evaluation-of-the

We addressed both comments by changing: *"We further adopted the scheme by introducing elevation classes, following Lipscomp et al. (2013). Calculating the SMB on fixed elevation classes, has the advantage that the model becomes computationally cheaper, as the SMB is computed on the native and coarse resolution atmospheric grid instead of the high-resolution ice sheet topography."*

Page 3, line 80: How is the lapse rate for these factors determined? Also, do you use the same lapse rate in summer and in winter? If yes, is this a valid assumption? What is the sensitivity to this choice?

The lapse rate for the corrections is constant over time and space, which is a caveat in our method. As we are using a global model it is challenging to incorporate different lapse rates for summer and winter, as it would require lapse rate values that depend on the location (e.g. southern and northern hemisphere). As we have a very different ice sheet configuration during the LGM as compared to present day we would not like to make variables location dependent. Also, it is not certain how these values would change for different climates and over different ice sheets. Here, we use values based on present-day estimates with a bias towards summer values (see also comment to Page 4, line 87).

Page 4, line 85: Not sure if I understand this description. How do you conserve water if this is the case?

Note, that we only use the MPI-ESM output as forcing for the EBM. Hence, we do not need to conserve water and it is irrelevant here. In a fully coupled simulation, where interactive ice sheets are included, this would lead to discrepancies but can be corrected through run-off.

Page 4, line 87: This lapse rate is quite a bit lower than the ICAO value of -6.5 K/km. Explain this choice, and did you test the sensitivity to this value?

The lapse rate is used in the model as tuning parameter and we have chosen a value that results in realistic SMBs for the present-day Greenland ice sheet. In the current version of the model it is considered as relatively low (4.6K/km), which certainly is at the lower end for Greenland temperatures. However, it has been shown that over ice sheets near-surface lapse rates are significantly lower than the ICAO values, specifically during summer (e.g. Gardener et al., 2007). As we only use one lapse rate for the entire year and over all ice sheets we have chosen a relatively low value that still lies within observational values.

Page 4, line 97: Technical detail but what happens to (latent and sensible) energy fluxes if the atmosphere model simulates liquid precipitation, but the height corrected temperature is below freezing (and vice versa)?

The energy fluxes are calculated from the height corrected variables, so this should not be an issue here. We added *"Latent and sensible heat fluxes are parameterized and calculated from the height-corrected variables."*

Page 5, line 136: It is a bit clunky to define WE at the end of the sentence. Can this definition be moved earlier in the sentence?
Thanks for pointing this out. We moved this part of the sentence further up.

Page 5, line 141: brighten -> increase (?)
Thanks, we changed this.

Page 6, line 165: "Once: : :" — Meaning here is not clear. Do you mean that it is removed?
We rephrased the entire paragraph for clarity according to the comments of reviewer #1.

Page 6, line 180: Technical detail, but please specify that this is the horizontal resolution
Changed.

Page 7, line 189: Acronym "ka BP" is not defined
According to the comments of the first reviewer we changed all the year definitions throughout the entire manuscript.

Page 7, line 195: Not sure if I fully understand this modeling strategy. Do you run 10 years with a constant forcing before updating the boundary conditions, or do you advance the orbital clock, topography, etc. by 10 years every model year?
The first! In our setup the model is run synchronous in time but the forcing fields are updated only every 10 years. We changed 'updated' to 'prescribed' to clarify.

Page 7, line 196: What happens to vegetation in areas that are deglaciated?
We added a sentence to explain this: "*Land cells that are deglaciated are covered with the same vegetation form as the adjacent grid cells.*"

Page 7, line 203: "it is a good proxy" –> "hence, it is a proxy for..."
We changed this.

Page 8, line 214: " for a long enough adjustment" is a bit colloquial
We changed to 'sufficient'.

Page 8, line 232: Why did you use ERA-Interim instead of the newer ERA5? Do you expect different results with a newer reanalysis product?
We used ERA-Interim, as the regional models used for comparison here are all forced with ERA-Interim. Using the same background climate allows us to assess the uncertainties due to the downscaling techniques (regional modeling vs. EBM_ERAI). The derived SMB can only be as good as the forcing. As observations and assimilation over the Arctic regions are still sparse we do not expect a significantly better climate for another reanalysis product.

Page 9, line 250: : : :historical climate conditions -> recent past
See earlier comment.

Page 9, line 254: remind the reader that this is the "coarse" and "low resolution" simulations
We added a reminder.

Page 9, line 264: Say something about how these numbers change over Greenland (latitudes of interest). Should be a factor of 2-3 difference from the equator
Thank you - it is a good idea to write the values for Greenland instead.

Page 10, line 286: Good reference https://tc.copernicus.org/articles/13/1547/2019/
Thanks for this reference. We added this reference and changed the paragraph according to the suggestions by reviewer #1.

Page 10, line 287: Sentence starting "The overestimation" is a bit clunky and can be simplified
Changed.

Page 11, line 316: What figure(s) are you discussion here?
We added a reference to Fig. 1 and 2.

Page 12, line 344: Good place to cite some work on atmospheric dynamics /circulation in the deglaciation. See major comment
Thanks, we did that! See comments above.

Page 12, line 354: shrinks -> recedes
Thanks! Changed.

Page 12, line 357: Further, it points towards the fact -> Further, it suggests
Thanks.

Page 13, line 380: typo: his -> its (?)
Yes.

Page 14, line 416: Even though the AMOC probably plays an important role for this response, it is the atmosphere that primarily interacts with the ice sheets. I would suggest extending this discussion with changes in the atmospheric circulation in mind, and perhaps cite a few papers that have looked at these interactions before
This is true, but we do believe that the AMOC changes are the trigger. Slowdowns of the AMOC lead to a significant cooling over the North Atlantic and the adjacent regions. Hence, they drive the changes in the surface temperatures that affect the SMB changes. All of this interaction is of course not possible without changes in the atmosphere. We tried to clarify this chain of processes and added references. We specifically added *"Another possible contributing factor to the pronounced SMB and ELA variability over the northern hemispheric ice sheets during this time period are changes in the atmospheric circulation. Lofverstrom et al. (2017) found that elevation changes of the North American ice sheet around the saddle collapse, defined by the separation of the Laurentide and Cordilleran ice sheets, caused significant changes in the stationary wave patterns. An amplifying factor for atmospheric circulation changes is the southward extension of the sea-ice cover due to the AMOC slowdown and reduced North Atlantic sea-surface temperatures. Such changes have a significant influence on downstream precipitation, evaporation and temperature patterns over the North Atlantic and adjacent areas."* in the end of this paragraph.

Page 15, line 467: What about regional and large scale atmospheric circulation?
We added feedbacks due to the ice sheet height, which includes atmospheric circulation changes.

Page 15, line 468: You could cite this paper when talking about other feedback processes: https://agupubs.onlinelibrary.wiley.com/doi/full/10.1029/2018RG000600
Thanks, we added this reference here.

Page 16, line 491: Can you end with a slightly more comprehensive future outlook? Where do you want to take this in the future, and how will this new modeling capability be used in e.g. simulations of the future climate evolution, and/or other paleo-climate states?
We added some information of the future path of our research here: *"Utilizing the SMB data set presented here as forcing for ice sheet model simulations will allow for an investigation of ice sheet dynamics during the last deglaciation. In the future, we will utilize the EBM in simulations with an interactive ice sheet model, which is currently employed within the MPI-*

*ESM setup in the scope of the project PalMod (Latif et al., 2016, Ziemen et al. 2019). This will allow to investigate feedback processes between ice sheets and the other climate components (see e.g. Fyke et al., 2018, for a recent review). It will also allow to investigate processes and test hypotheses arising from the deglaciation simulations for other climates, such as e.g. the last glacial inception, Marine Isotope Stage 3 as well as the future."*

Table 2: This simulation contributed to CMIP6
Thanks. We changed this.

Figure 1: Here and elsewhere. Spectral colorbars are bad for people with color blindness. Please us a non-spectral color scale if possible
We have revised all figures and changed the colorbar.

Figure 4: Here and elsewhere. The SMB colorscale is a but crowded. If possible, use fewer intervals.
We have revised all figures and changed the labels, according the ones used in van Kampenhout et al., 2019.

---

## Author Response (AR2)

**Response to the review of the manuscript "Analysis of the Surface Mass Balance for Deglacial Climate Simulations" submitted to The Cryosphere.**

We thank the reviewer for reviewing our manuscript a second time and his detailed explanations where and how to improve the manuscript. In the following, reviewer comments are highlighted in blue, author responses in black.

**Anonymous Referee #1**

**General comments**

The responses to initial review comments are all fine, and I think the paper has been improved as a result. There were a still a number of parts of the model description that I still found unclear, especially the albedo and density evolution sections - clarifying those further would be the main thing I would suggest at this point. I was reasonably happy enough with what science analysis there is of the simulations in my first review, I've not really added anything additional to what I said first time on that part of the paper.

We have addressed all the comments and specifically tried to clarify the section including the albedo and density evolution.

**Detailed comments:**

line 5: add "northern hemisphere ice sheets" into "for the last deglaciation"?

We changed the sentence to 'study changes in the SMB and equilibrium line altitude (ELA) for the Northern Hemisphere ice sheets throughout the last deglaciation'.

line 6: it's not clear here what this spatial resolution is higher than

We added 'The EBM is used to calculate and downscale the SMB on higher spatial resolution than the native ESM grid and...'.

line 13: the meaning of "SMB/ELA" might be clearer as "SMB and ELA"

Thanks, we changed this.

line 26: there are RCM studies for future ice sheet SMB too of course, eg Fettweis et al 2013 doi:10.5194/tc-7-469-2013

This is of course true and these studies were not meant to be excluded here! We revised the sentence to 'as well as simulations with high-resolution regional climate models (RCMs), which are constrained by reanalysis or ESM data at the lateral boundaries, and cover the last century and near future only' and added the reference. We also added this information to the abstract.

line 42: there's inconsistent capitalisation of Northern Hemisphere here (and elsewhere?), compare line 39

Thanks for pointing it out. We have unified it through the entire manuscript.

line 53: this last sentence is rather long and unwieldy. It's not obvious there's a direct "resemblance" between the deglacial climate changes just described and "expected future climate change"

Thanks for pointing this out. We changed the sentence as following 'The significant climate changes and variability associated with the changes in the Northern Hemisphere ice sheets emphasize the need for a realistic representation of the SMB for past and future stand-alone ice sheet and coupled climate-ice sheet model simulations (Fyke et al., 2018).' We do believe that this argument stands for itself. The resemblance to the future in terms of the rate of sea-level rise was mentioned earlier in the paragraph.

line 62: (and other places the future study is referred to) can this be cited as "Author et al, (in prep)" or similar, to give future readers a clue how to go looking for this companion paper in the literature when it has been published.

We added '(*Kapsch et al., in preparation*)' in all places that we refer to this study but are not fully aware how it is handled by the journal.

line 84: further "adapted" the scheme, perhaps? I'm not sure these sentences would adequately explain what elevation classes/levels are to a reader that didn't already know. If they are not clear on that it then makes it very confusing to then be told, having started the paragraph by saying the EBM downscales melt and accumulation onto the high resolution ice sheet topography, that you now \*want\* to calculate SMB on the coarse resolution atmosphere grid.

This might indeed be confusing. We changed the sentences to 'We further adapted the scheme by introducing elevation classes, following Lipscomb et al. (2013). Calculating the SMB on fixed elevation levels, has the advantage that the model becomes computationally cheaper and that the obtained 3-D fields can be interpolated onto different ice sheet topographies (see Section 2.3).' We believe that people not familiar with elevation classes can refer to Section 2.3 or Lipscomb et al. 2013.

**line 113: I didn't understand "is accumulated" before "and falls as snow"**

We changed the paragraph slightly to 'Accumulation and melt determine the SMB. Accumulation is controlled by precipitation and takes place if precipitation falls as snow. In the EBM precipitation is considered as snow with a density of 300 kgm-3 when the heightcorrected near-surface air temperature is lower than the freezing temperature of 273.15 K. Otherwise, precipitation falls as rain.'

line 118: "includes", instead of "consists of", since the SBM does more than just snow layer modelling?

Thanks for the suggestion.

line 123: "fluxes are parameterised" - if you say this, do you need to say how?

As we use standard bulk aerodynamic formulas we changed the sentence to 'Latent and sensible heat fluxes are calculated from the height-corrected variables using bulk aerodynamic formulas' and believe it is not necessary to include formulas.

line 127: you've given what look like one end-member of the density-dependent conductivity, it would be clearer to say what the other is too perhaps

We have removed part of the sentence, as the heat conductivity is calculated following the density, hence, changes over time. We agree with the reviewer that giving one end-member does not give more insight on the function itself without an additional equation (see reference to Schwerdtfeger, 1963).

line 130: "observations" needs a reference

We added a reference.

line 152: can a darker background "shine" through? "Show", maybe.

Yes, that is a good suggestion. Thanks.

line 156: why does snowfall only increase thickness when it's snowing faster than a certain rate?

We have chosen a rate dependence to allow for the fact that only a closed snow cover will alter the albedo. E.g. if little snow falls in an ablation area it will melt almost immediately. To account for this we chose this threshold.

line 157: "all depths presented here [..]" is repeated from line 144

Thanks, an over left of the last revision.

line 161: add "surfaces" to "melting and refreezing have different albedo values". I still don't understand the description of the refreezing albedo evolution, I'm afraid. A surface that is simply accumulating snow has an albedo set by eqs 1 and 2, right? If the temperature rises and the surface starts melting, it gets given the constant fixed value of alpha\_{snowmelt} or alpha\_{icemelt} (line 162) - what determines which of these is used, a density threshold in the top layer? If temperatures drop and the surface can refreeze, the albedo jumps up to alpha\_{snowrefrz} or alpha{ice\_refrz} - this time you do say the choice depends on the snowdepth (the d\_{snow} defined on line 152?), but not how. Aging is now said to start again, I assume using eq1 with alpha\_{Xrefrz} instead of alpha\_{frsnow} - but since the alpha\_{Xrefrz}s are lower than alpha\_firn}, does aging now make the surface brighter again as eq1 pulls alpha\_{snow} toward alpha\_{firn}, or is a different target albedo used?

If the snow depth is larger than a threshold alpha snowmelt is chosen, otherwise we use alpha icemelt. We have revised the paragraph accordingly. To the last question: this is correct. Refreezing starts again with Eq. 1 in which we replace alpha\_frsnow with alpha\_refrz for snow and ice. Alpha\_firn is replaced by alpha\_refrzold, which is defined as alpha refrzold= $(2 \times alpha refrz + alpha melt)/3$ . So we are aiming for a different target albedo here than simply firn. In the case of ice the target albedo lies by 0.55, in case of snow by 0.63. Hence, the surface does not get brighter but darker, as we are aiming for another target albedo as alpha\_firn. Accordingly we changed the text to 'Depending on the snow depth, melting and refreezing surfaces have different albedo values (Table 1). If the snow depth lies above or below a snow depth threshold of 0.25 cm the albedos for snow and ice are used, respectively. When the surface experiences melt, the albedo drops to the albedo of snow or ice melt (alpha {snowmelt/icemelt). When the surface refreezes, the albedo potentially increases (alpha snowrefrz or alpha icerefrz) and the aging process starts. The aging process is similar to that for snow processes as described in Eq.~1, but the albedos for refrozen surfaces are smaller (alpha snowrefrz/icerefrz, the reference albedo alpha firn reduces to alpha\_refrz = (2 alpha\_snowrfz/icerefrz+alpha\_snowmelt/icemelt)/3 for refrozen snow or ice, respectively). Additionally, the process is slower (tau ar). Only melted surfaces and the background do not experience any aging.'

line 171: I found this phrasing confusing - cloud cover can't physically affect the surface reflectivity itself, you let it do so in your scheme to compensate for a lack of spectral resolution. "Furthermore, as part of our broadband albedo parameterisation we let varying cloud cover affect [...]"

**Thanks for this good suggestion.**

line 180: I still don't understand the description of the density evolution after ablation, I'm afraid! I may have a completely incorrect paradigm in my head for how your snow pack works. To take an extreme example: say there's been an extended period of ablation. Density profiles are shifted upwards, "an inflow of ice through the bottom layer closes the mass budget". Does the total mass in the snowpack remain the same, with the same number of layers (of different thickness), or do you lose layers entirely? If the influx of ice at the bottom means you keep the same number of layers in your snowpack, but shift the densities everything has, you end up with an entire set of layers at the density of solid ice. What happens to the density profile when fresh snow starts to fall on this case? The accumulation paragraph above only talks about increasing a layer's density until it reaches the appropriate reference density - but if you still have all the layers, just at ice density, the top layer is already above this reference state. Do you actually lose layers, and then fresh snow starts making new ones on top? This all comes back to my question in the first review that you give mechanisms for increasing density above the target/reference profile, but none that look like they can reduce density again.

We thank he reviewer for his feedback to this paragraph and his examples, which helped to identify the 'missing pieces' in this section. We have entirely revised this section in order to address the concerns raised by the reviewer. One cause for the misunderstanding is probably the lack of explaining how the vertical advection of mass works within the model, which we have now included. The revised paragraph now reads as following 'For a non-zero SMB, the thickness of the uppermost layer of the snow model would change. To compensate for that, a simple 1-d advection scheme conserving heat and mass is applied. In case of surface ablation, the densities and temperatures are advected upward. As lower boundary condition, we assume an inflow of ice with a density of 917kgm^-3 and a temperature of the lowest model layer. In case of accumulation, an inflow of snow with a density of 300 kgm^-3 and a temperature of the height corrected near-surface air temperature is assumed as upper boundary condition. To account for snow compaction we introduce the aforementioned reference density profile Cuffey, (2010). If the downward advected snow into a layer is smaller than the reference density in that layer the density of the in flowing snow is set to the reference density and the flow to the layer below is reduced accordingly to conserve mass. Once the reference density is reached in all layers, mass flows out of the bottom layer and is removed from the system. As a consequence of this procedure the density in each layer lies always between the reference density (in case of permanent snow accumulation) and the density of pure ice (in case of permanent ablation)'.

**line 197: "Pfeffer" citation missing a year**

**Thanks!**

line 224: Is JSBACH is allowed to interactively evolve the vegetation distribution once the ice sheet has receeded from a grid box, or do you stick with whatever is prescribed from the neighbouring cells when the ice disappears?

If the ice sheets recedes from a grid cell the vegetation is set to bare soil. It can then evolve freely depending on the climate within JSBACH. We changed the sentence to clarify this to "Land points that are deglaciated are covered with bare soil Ocean cells that become land due to changes in the sea level are initialized with the same vegetation form as the adjacent grid cells while. After this initialization the vegetation of the grid cells evolves interactively within the dynamical vegetation model JSBACH."

line 392: I think "should not always" would be better than simply "cannot" here.

**Thanks, we changed this.**

line 526: I'd start a new paragraph at "Utilizing"

**Good suggestion.**

Table 1: the comments on alpha\_{bg} refer to "snow aging, Equation 1", but alpha\_{bg} is used in equation 2, and does not seem directly related to aging

This is a mistake. We changed this and refer to Equation 2, which refers to the modulation of the surface albedo due to the background albedo, as well as Equation 3 for the definition of alpha\_bg. Thanks for pointing this out.

---

## Author Response (AR3)

Dear Harry,
Thanks a lot for your suggestions. Thanks a lot for all the suggestions. We are glad, that the manuscript has improved and are very glad for all the input! Please find a point-to-point reply to your suggestions below.

l. 9-11: "...The increase in insolation and associated warming early in the deglaciation result in an ELA and SMB increase. The SMB increase is caused by compensating effects of melt and accumulation, as a warmer atmosphere precipitates more?": this is an intriguing finding, which makes it quite difficult to understand for the reader (especially at this stage in the abstract). When you speak about 'compensating' effect, I would expect both processes kind out keeping each other in balance. But my understanding here is that the increase in precipitation outweighs the increase in melt overall (due to which the SMB increases)? And given that the ELA moves to higher elevation, do I understand it correctly that the increase in SMB mainly occurs at high elevation, while at lower elevation the SMB decreased (to a lesser extent)? As you explain elaborately in lines 396-392, this whole process is rather "counter-intuitive". It is therefore important to be really clear in the introduction, to not immediately raise questions related to this finding. May be worth to rephrase to something like: the SMB increase occurs at high elevation (due to increased precipitation) and this effect outweighs the decrease in SMB that occurs at low elevation and causes the ELA to rise.
Thanks for pointing this out. We have rephrased the sentence to "The SMB increase is caused by compensating effects of melt and accumulation: the warming of the atmosphere leads to an increase in melt at low elevations while it results in an increase in accumulation at higher levels, as a warmer atmosphere precipitates more." to make the effect more clear.

Several occurrences where I had to read the sentence several times before understanding it. Here could likely be solved by adding a "," at the correct location, e.g.:
• l.30-31: had to read the sentence several times before I understood it: suggest adding a ',': "...and climate change, output from..."
• l.69: "... long-term climate simulations the coupled..."    "long-term climate simulations, the coupled..."
• l.220: "For grid cells over land melt water..."    "For grid cells over land, melt water..."
Thank you for the careful reading. We added the commas in the proposed locations.

l.64-65: "A thorough evaluation of the long-term model simulations and their forcing data sets used here is subject to a future study (Kapsch et al., in preparation)": here, at the end of the introduction you start referring to future work, which you did not perform here yet. Would recommend leaving this out of introduction. A related question: in the last sentence you refer to a dataset that will be made available to the community: does this refer to the work you present here (if so, refer to 'this dataset is made available'), or also to future work you plan to perform (if so, also best to remove this from the introduction).
We added these references in the introduction in response to previous reviewer comments. We do not have a strong opinion on that, so we removed the reference here in the introduction. Regarding the data set. We changed 'will be made available' to 'is available to the ice-sheet community', as the data set is publicly available and referred to in 'Code and availability'.

In section 2.1, the subdivision of the subsections is not entirely clear: some subsections are not numbered but presented in bold ('height correction' and 'surface mass and energy

balance calculation'), or given in italic bold ('Albedo' and 'Vertical advection and density evolution'). Would recommend numbering all the subsections (potentially introducing new subsections also), otherwise difficult to follow what belongs to what. Another option could consist of removing the all subheading without numbering. Same goes for the non-numbered sections in 2.3 and in section 3.

We changed the headings for better consistency. Specifically, we renamed some sections and chose to number only second order labels.

l.234-235: "we branched off a last millennium simulation at 950 a BP (years before present)": without any explanation, the choice for branching off 1000 years before 1950 seems arbitrary. Why is the branching off for instance not happening 2 or 0.5 ka before 1950? Could you provide a short explanation (one or two sentences should do) why the branching off occurs over last millennium? (maybe a common approach in your field? If so, would then be good to specify this for a non-expert)

We added one general sentence and a reference to last millennium simulations: 'Within PMIP, last millennium simulations are used to provide a seamless transition between the last deglaciation and the historical period (Junglaus et al., 2017). For the simulation, topography, land-sea and glacier masks, [...]'. We reasoned further down in the paragraph also why we have chosen the millennial forcing 'Overall, the applied forcing allows for a more realistic treatment of atmospheric processes associated with changes in e.g., ozone, aerosols, $CO_2$ concentrations, and land use, and it accounts for their climatic impacts for present day climate conditions.' The millennium simulations allow the model to adjust to the changes in the forcing, which we mention also in the paragraph 'For the evaluation only the years 1980-2010 are used, which allows for a sufficient adjustment of the model to changes in the forcing.'

l.260: "(see Section 2.3 Nowicki et al., 2016; Fettweis et al., 2020)"  "(see Section 2.3; Nowicki et al., 2016; Fettweis et al., 2020)"

Thanks. We changed this.

l.261: in this section, you explain that for the evaluation you rely on ERA-Interim. However, since some time now, ERA5 is out, which is from many perspective a better product than ERA-Interim, which you describe as: "one of the best reanalysis products available" (l. 266-267). Any reason for using ERA-Interim (4th generation product) and not the newer ERA5 (5th generation product)? Especially because you use the new CMIP6 data, I was also expecting to have ERA5 for the past. But maybe this is because the time period available for ERA5 was problematic? Or because ERA5 was not available at the time of analysis (like you mention for instance in earlier section at one point in l.239: "For the years beyond 2010, the forcing fields in the desired resolution were not available at the time of the analysis"). Or because MAR was also forced with ERA-Interim? I am just guessing here, so would be good to shortly mention this in text / provide a brief explanation.

We chose ERA-Interim, as the regional models are using ERA-Interim as boundary forcing. We added a sentence to the section: 'ERA-Interim was chosen, as it is used as boundary forcing in the RCM simulations introduced in the end of this section, which are used for a more thorough evaluation.'

l.331: 'melt' section: for consistency reasons, would it be possible to make a short / qualitative comparison with RACMO here in the text, as you did for the precipitation by

referring to Noël et al. (2019): i.e. just include a sentence on the RACMO value that you give in Table 3, and how this compares to your model simulations.

We added a sentence at the beginning of the paragraph: 'Integrated over the ice sheet, EBM_ERAI, EBM_MPI-ESM-CR, and EBM_MPI-ESM-LR simulate less melt than MAR and RACMO (Table 3) but the sign of the differences varies significantly depending on the region (Fig. 1).'

l. 345-350: you explain some problems occurring with ERA-Interim related to e.g. an "unrealistic representation of cloud optical properties" and "low surface albedos". Out of curiosity: do you think this is better represented with ERA-5, and would this change the outcome of your comparison? If so, potentially worth adding a comment on this.

We have not looked into the ERA-5 reanalysis but the evaluation of reanalysis data is challenging, especially for these kind of parameters (measurements are sparse, specifically in the regions that we are most interested). As we have not mentioned the ERA-5 reanalysis we do not believe that this would add significantly. Also, our main focus is not the ERA-Interim reanalysis but the MPI-ESM-CR simulations. Hence, we did not add any sentence here.

l.353-355: long and difficult sentence to understand. Consider splitting the sentence in two to increase its readability.

Thanks for pointing this out. We have split the sentence to 'The evaluation shows that major differences between MAR and EBM_MPI-ESM-CR are the increased melt on the western flank of Greenland and along the coastal areas as well as the overestimation in accumulation in the southern part of Greenland. The latter can partly be reduced by increasing the model resolution, as shown by comparisons with EBM_MPI-ESM-LR.'

l. 403: "Although melt and accumulation again partly compensate, ...", do you mean that these two compensate one another? Maybe specify to be even clearer: "Although melt and accumulation again partly compensate one another,..."

Thanks for the suggestion. We changed this to 'compensate each other'.

l.414-422: here you describe the state of the ice sheet around the Holocene Thermal Maximum, but you do not explicitly refer to this period / this terminology. Any reason for this? If you think this is adequate, would support a short reference to this period, potentially including references to some studies/observations that have quantified the extent of the ice sheet during this period.

There is no reason that we did not include a reference to this period. We have included a sentence and referred to a recent review paper of this period. 'The minimum SMB (216 Gta^-1) is reached at 8.7 ka, and the maximum ELA (1556m) occurs at 9.3 ka (Fig. 3 and 4), corresponding to the Holocene Thermal Maximum (for a recent review see Axford et al, 2021).'

l.528: "In the future, we will utilize the EBM in...": well, one never knows 100% how things turn out in science and what will happen. Seems really interesting, and hope you can do this, but would suggest having a slightly more conservative/tentative wording: "In the future, we plan to utilize the EBM in..."

In fact, we already completed our first transient fully coupled climate-ice-sheet simulations for the last deglaciation. However, tuning of the model components is still ongoing and the work has not been published yet. ☺ So we take the suggestion despite our first success.

Thank you once again for submitting your work to TC and for fruitfully interacting with the two anonymous reviewers (please acknowledge their help in 'acknowledgments' section also).

Thanks a lot for reminding us! We are very grateful for the discussions with the reviewers and your final suggestions! We added 'Additionally, we are grateful to Thomas Kleinen and two anonymous reviewers for their helpful comments and discussions, which helped to significantly improve this manuscript.'

With best regards,
Marie